# Single-cell transcriptomics reveals the molecular basis of human iPS cell differentiation into ectodermal ocular lineages
Laura Howard [1,2], Yuki Ishikawa[3,4], Tomohiko Katayama[3,4], Sung-Joon Park [5], Matthew J. Hill [2], Derek J. Blake [2], Kohji Nishida[4,6] ✉, Ryuhei Hayashi [3,4] ✉ & Andrew J. Quantock[1]

The generation of a self-formed, ectodermal, autonomous multi-zone (SEAM) from human induced pluripotent stem cells (hiPSCs) offers a unique perspective to study the dynamics of ocular cell differentiation over time. Here, by utilising single-cell transcriptomics, we have (i) identified, (ii) molecularly characterised and (iii) ascertained the developmental trajectories of ectodermally-derived ocular cell populations which emerge within SEAMs as they form. Our analysis reveals interdependency between tissues of the early eye and delineates the sequential formation and maturation of distinct cell types over a 12-week period. We demonstrate a progression from pluripotency through to tissue specification and differentiation which encompasses both surface ectodermal and neuroectodermal ocular lineages and the generation of iPSC-derived components of the developing cornea, conjunctiva, lens, and retina. Our findings not only advance the understanding of ocular development in a stem cell-based system of human origin, but also establish a robust methodological paradigm for exploring cellular and molecular dynamics during SEAM formation at single-cell resolution and highlight the potential of hiPSC-derived systems as powerful platforms for modelling human eye development and disease.

The generation of a self-formed, ectodermal, autonomous multi-zone (SEAM) relies on intrinsic developmental processes to guide the differentiation and self-organisation of human induced pluripotent stem cells (hiPSCs) into ocular tissues[1,2]. A growing two-dimensional SEAM primordium can be identified according to the progressive delineation of four concentric zones, with cellular location being indicative of lineage, which recapitulates characteristics of whole-eye morphogenesis. An in-depth characterisation of maturing ocular SEAMs has revealed spatial patterning centred around an innermost zone, Zone 1, which is enriched with cells expressing neural markers indicative of presumptive neuroectoderm. Extending radially, cells in Zone 2 closely resemble those of the neural retina and retinal pigment epithelium (RPE), while Zone 3 is predominantly occupied by ocular epithelial cells. *PAX6*, a master regulator for eye development[3,4] is expressed in cells across Zones 1 to 3. In

contrast, at the periphery of the SEAM, Zone 4 contains cells which express epithelial markers in the absence of *PAX6* and likely represents general surface ectoderm. Functionally, SEAM-derived corneal epithelial cells have been isolated and expanded into sheets and these can successfully recover function when transplanted onto experimental ocular wounds[1,5]. Ocular surface epithelial cells isolated from Zone 3 of the SEAM have also been grown under defined conditions to form 3-dimensional organoids which are strikingly similar to native lacrimal glands. These organoids contain specialised cellular subtypes such as acinar and ductal cells, and when transplanted adjacent to the eyes of recipient rats adopt many of the defining features of functional lacrimal glands[6].

Since their discovery[7], induced pluripotent stem cells have garnered significant attention owing to their unique ability to differentiate into

[1]School of Optometry and Vision Sciences, Cardiff University, Cardiff, Wales, UK. [2]Centre for Neuropsychiatric Genetics and Genomics, School of Medicine, Cardiff University, Cardiff, Wales, UK. [3]Department of Stem Cells and Applied Medicine, Osaka University Graduate School of Medicine, Osaka, Japan. [4]Department of Ophthalmology, Osaka University Graduate School of Medicine, Osaka, Japan. [5]Institute of Medical Science, University of Tokyo, Tokyo, Japan. [6]Institute for Open and Transdisciplinary Research Initiatives, Osaka University, Osaka, Japan. ✉e-mail: knishida@ophthal.med.osaka-u.ac.jp; ryuhei.hayashi@ophthal.med.osaka-u.ac.jp

virtually any cell type, making them a powerful tool for both disease modelling and regenerative medicine. Meanwhile, the emergence of unbiased single-cell transcriptome-wide analyses has revealed unprecedented levels of phenotypic and functional heterogeneity amongst populations of cells, and this new knowledge has far-reaching implications in both basic and translational research. In the field of ophthalmology, the creation of hiPSC-derived cellular structures for transplant surgery is an emerging possibility. This is particularly significant given both the need to develop minimally invasive surgical techniques and the scarcity of donor tissue worldwide. Here, we use single-cell analyses to interrogate developing SEAMs at key junctures throughout their formation as proxies of human eye development, in order to reveal the developmental trajectories of individual cell populations as they lose pluripotency and commit to specific ocular lineages.

## Results

### Initial gene expression and cell lineage identity in the formative SEAM; weeks 0-1

Human iPSCs were initially grown for 10 days in StemFit medium to preserve pluripotency before being transferred into differentiation medium in order to promote SEAM formation. Cultures were closely monitored to ensure the appearance of the characteristic zones shown in Fig. 1a–c, after which SEAMs were harvested as described previously[1,2]. Single-cell RNA sequencing was performed using the 10x Genomics Chromium platform. Following extensive QC, data were normalised using SCTransform v2[8] and clustered using the in-built Seurat[9] function 'FindClusters', which returned 12 populations (Fig. 1d). Gene expression markers for all identity classes were extracted using the FindAllMarkers function. Notably, cells at week 0 (WK0) were highly proliferative (Supplementary Fig. 1) and strongly expressed key pluripotency markers including *POU5F1* (*OCT4*)[10,11], *ESRG*[12], *MIR302CHG*[13] and *DPPA4*[14] (Fig. 1e). *SOX2*, which functions by forming a complex with POU5F1 and activating or repressing the expression of various target genes which are required for maintaining the undifferentiated states of pluripotent cells[15,16], was also expressed ubiquitously at WK0, while *NANOG* was expressed by only a subset of cells (Fig. 1e).

The shift from a pluripotent state to a state of lineage commitment is marked by significant fluctuations in gene expression and the progressive weakening of pluripotent stem cell networks. During vertebrate eye development, so-called eye-field transcription factors are expressed in overlapping domains, likely beginning with the onset of expression of *OTX2* in the anterior neural plate, with the eye-field transcription factors *PAX6*, *SIX3*, *SIX6*, *RAX* and *LHX2* then becoming detectable in the presumptive eye-field[17,18]. With this in mind, of particular interest for our analyses at this stage are clusters 7, 8, and 9 (Fig. 1d). Cells in these clusters expressed ectodermal and eye-field transcription factors, whilst concurrently displaying reduced expression of pluripotency markers (Fig. 1e, f). This suggests that the intrinsic developmental pathways pertaining to anterior ectodermal specification are already underway in those cells in which pluripotency is diminishing.

After culture for 1 week in differentiation medium (WK1) (Fig. 1g), there was significant and progressive downregulation of several key pluripotency markers (Supplementary Fig. 2), and cluster analysis revealed early divergence of cellular identity towards either surface ectodermal or neuroectodermal fates (Fig. 1h). While some cells retained expression of pluripotency markers (e.g. *POU5F1*), keratin family members and surface ectoderm markers such as *CLDN4*, *CLDN7*, *CDH1*, and *EPCAM* were strongly expressed in clusters 5, 6 and 10 (Fig. 1i). *BMP4* was also strongly expressed by cells in clusters 5 and 6 (80.16 and 86.89% of cells, respectively) and is widely known to play a crucial role in promoting ectodermal-epithelial cell lineages and inhibiting neural fate[19,20]. *DLX5*, a downstream effector of BMP4 in non-neural ectoderm[21], again showed robust expression in these cells (Fig. 1i). At this stage in the differentiation protocol, the cells can be described as forming an immature 'pre-SEAM'. Ocular surface ectoderm cells in the pre-SEAM which display corneal epithelial commitment express *PAX6*, whereas *PAX6*-negative cells are likely to represent early general (head) surface ectoderm[1]. In our data, this is clearly illustrated by the strong expression of *PAX6* in cluster 6, but not cluster 5 (Fig. 1i). Cells in cluster 10 were also *PAX6*-negative, likely representing their status as pluripotent cells which are newly differentiating towards surface ectodermal fates, but before the establishment of ocular vs. non-ocular commitment.

As the pre-SEAM continues to grow and mature in culture, there is evidence to suggest the emergence of satellite spheres containing *SOX10 + / p75+* neural crest cells[1,2]. The periocular mesenchyme, which is derived from neural crest cells which migrate into the ocular region during development, gives rise to cells of the corneal stroma and endothelium[22]. Several key transcription factors that are known to be important for corneal development are expressed in the periocular mesenchyme. These include *PITX2* and *FOXC1*[23], along with *TFAP2B*, which is thought to be required for both epithelial stratification and endothelial differentiation[22]. In our analysis, we find that a subset of cells within the epithelial clusters 5 and 6 express markers of cranial neural crest/periocular mesenchyme, including *NGFR*, *PITX2*, *FOXC1/2*, *SOX10* and *TFAP2B* (Fig. 1j). Given that formation of the cornea depends on complex bi-directional interactions between the head ectoderm and periocular mesenchyme[24], it is possible that these reciprocal events are important for corneal development within the growing SEAM. The remaining cells at WK1 formed a large supercluster comprised of clusters 0-4, 7-9 & 11-12 (Fig. 1h). These cells were transcriptionally similar and expressed *SOX2*, which marks progenitors, alongside early neuroectodermal markers including *OTX2* and *CDH2*. The classical ocular neuroepithelial markers *PAX6*, *RAX*, *SIX3* and *LHX2* were also expressed by cells in these clusters, which are quite distinct from cells in the presumptive surface epithelial clusters expressing *EPCAM* and *CDH1*, and the pluripotent cells expressing *POU5F1* (Fig. 1k).

### Early gene expression and cell lineage identity in emerging SEAMs; weeks 2 & 4

During eye development, contact of the distal portion of the optic vesicle with the surface head ectoderm results in the formation of the lens placode. Subsequently, it is the invagination of this non-neural placode that generates a 3-dimensional structure, comprising the lens vesicle and optic cup. We began our analysis of 2-week-old SEAMs (WK2; Fig. 2a, b) by assessing the expression of genes reported to be involved in vertebrate lens development. Cells in clusters 1, 8 and 9 were *PAX6*-positive and expressed early lens markers including *FOXE3*, *PITX3* and *PDGFRA*[25–27] along with the developmentally regulated enzyme crystallin *BHMT*[28] and *CRYAB*, which interacts with other crystallins to help preserve lens transparency[29] (Fig. 2c). Alongside the eye-field transcription factors *PAX6*, *RAX* and *SIX6*, cells within clusters 1, 3, 8, and 9 also expressed *VSX2*, a well-established marker of multipotent retinal progenitor cells (RPCs)[30,31] and the retinal pigment epithelial (RPE) determinant *MITF*[32,33] (Fig. 2d), indicating a mixed pool of progenitors.

Five clusters of proliferative cells were present in the WK2 data (clusters 5, 7, 8, 9 and 11). These cells expressed markers such as *TOP2A*, *MKI67* and *UB2EC*, and cell cycle analysis demonstrated that a considerable percentage of these cells were in the S and/or G2/M phases (Supplementary Fig. 1). While cells in clusters 8 and 9 co-expressed lens and retinal progenitor cell markers, the proliferative cells of clusters 5 and 7 were transcriptionally more similar to those in cluster 0, with expression patterns indicative of progression towards a neural fate. We observed increased levels of the anterior neuroectoderm and developing forebrain marker *FOXG1* in these cells, along with *TTYH1*, which enhances NOTCH signalling and is required for the maintenance of neural stem cells[34], *PTPRZ1*, which is expressed by oligodendrocyte precursors[35], and the proteolipid *PLP1*, which is expressed in myelin[36] (Fig. 2e). These gene expression patterns are therefore indicative of a mixed neural and glial precursor cell population. Another NOTCH-responsive factor, *HES5*, which can regulate both gliogenesis and neurogenesis[37], was expressed across these clusters, whereas the pro-neural *HES6* was expressed in cells of cluster 10, together with neuronal markers such as *TUBB3* and *DCX* (Fig. 2e).

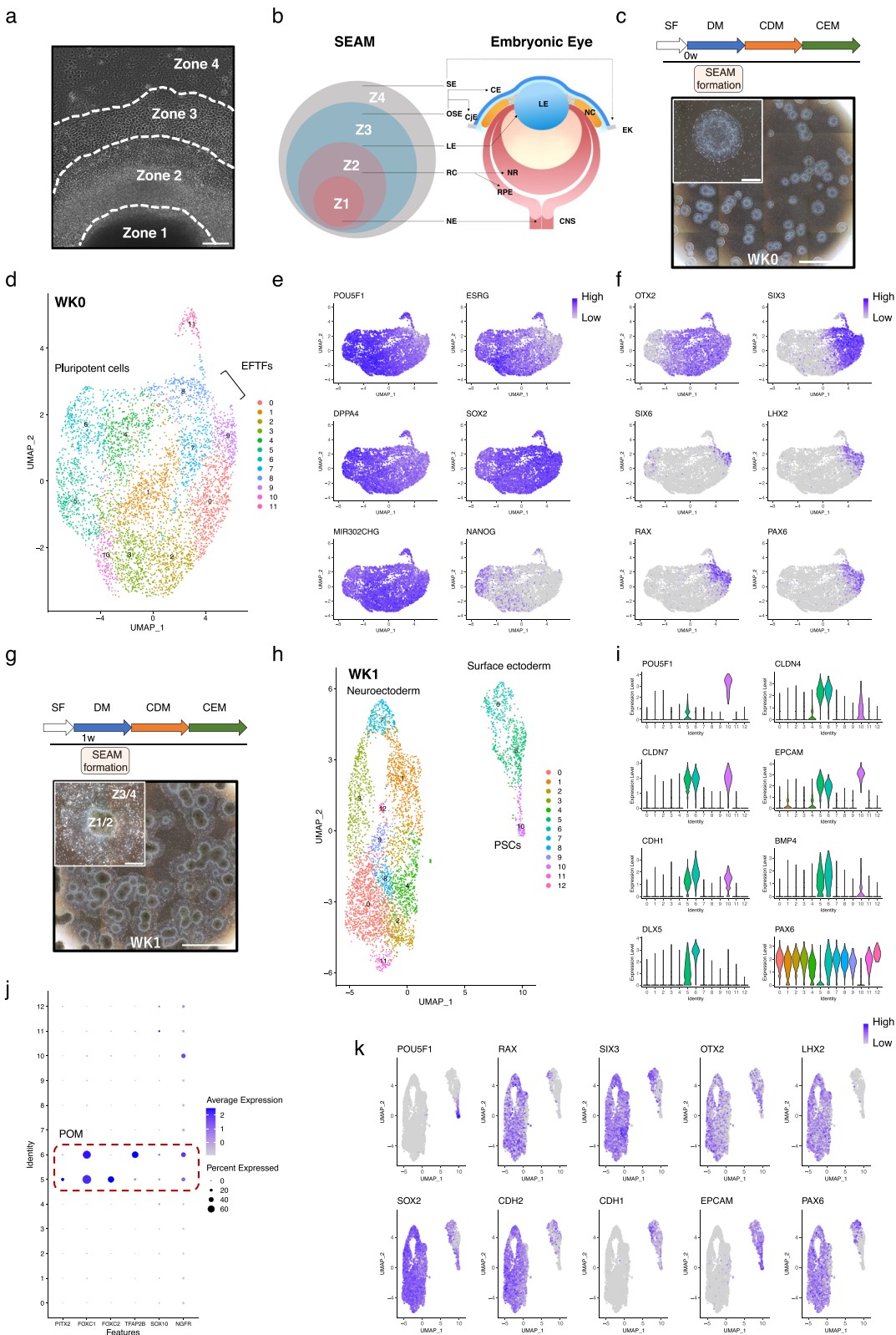

The remaining four clusters at WK2 (clusters 2, 4, 6 and 11) include subpopulations of cells expressing *POU5F1* and the stem cell marker *ABCG2*[38,39], and clusters representing developing surface ectoderm (Fig. 2f). Cells in cluster 6 expressed PAX6 along with the keratin family members *KRT8* and *KRT18* and epithelial markers including *EPCAM* and *CDH1*. *TP63*, which regulates epithelial cell stratification in the developing limbus

and cornea[40,41], was expressed by a small number of cells at this stage (Fig. 2f). Cells in cluster 2 were transcriptionally similar to those in cluster 6, but expressed little or no *PAX6*, indicating that these represent non-ocular epithelium, while those in cluster 11 expressed proliferative markers (Supplementary Fig. 1). Members of the claudin and S100A multi-gene families, including *CLDN4* and *S100A11*, which are known to be expressed in ocular

**Fig. 1 | scRNAseq analysis of WK0 and WK1 SEAMs. a** Phase-contrast image of a typical differentiating SEAM after 40 days of culture, showing the four characteristic zones. Scale bar: 100 μm. **b** Schematic depicting relationship of each zone to the developing embryonic eye. **c** Experimental timeline and representative phase-contrast image showing WK0 SEAMs in culture. Scale bar: 5000 μm; inset, 500 μm. **d** UMAP plot of WK0 SEAMs. **e** Feature plots showing expression of pluripotency markers. **f** Feature plots showing expression of early markers of ocular differentiation. **g** Experimental timeline and representative phase-contrast image showing WK1 SEAMs in culture. Scale bar: 5000 μm; inset, 500 μm. **h** UMAP plot of WK1 SEAMs. **i** Violin plots showing restriction of pluripotent gene expression (*POU5F1*)

and expression of surface ectoderm markers in clusters 5 and 6. **j** Dot plot showing co-expression of CNN / POM markers by cells in cluster 5 and 6. **k** Feature plots showing segregation of pluripotent, neuroectodermal and surface ectodermal cells in WK1 SEAMs. Z1 Zone 1, Z2 Zone 2, Z3 Zone 3, Z4 Zone 4, SF StemFit medium, DM differentiation medium, CDM corneal differentiation medium, CEM corneal epithelium maintenance medium, EFTFs eye-field transcription factors, NE neuroectoderm, NC neural crest, RC retinal cells, RPE retinal pigment epithelium, NR neural retina, LE lens, OSE ocular surface ectoderm, SE surface ectoderm, CE corneal epithelium, CjE conjunctival epithelium, EK epidermal keratinocyte, PSC pluripotent stem cell, CNN cranial neural crest, POM periocular mesenchyme.

tissues[42,43], were widely expressed amongst the epithelial populations at WK2, with markers of periocular mesenchyme including *TFAP2B* also present in subsets of cells interspersed within the main clusters (Fig. 2f).

At WK4 (Fig. 2g, h) there was widespread, multi-cluster expression of neuroepithelial and retinal progenitor cell markers including *RAX, SIX6, SOX2, VSX2,* and *SFRP2* (Fig. 2i) and at this developmental timepoint no pluripotent cells were detected (Supplementary Fig. 2). Cells in cluster 3 expressed *MITF, DCT* and *PMEL*, markers associated with melanin biosynthesis in the RPE[44,45], in regions where *VSX2* expression was lower, denoting progressive delineation of the presumptive RPE and neural retina (Fig. 2i). Highly proliferative progenitors mark cluster 6 (Supplementary Fig. 1), while cells in cluster 12 represent those of the developing lens. FindAllMarkers revealed that compared to other clusters, cells in cluster 12 differentially expressed *FOXE3, CRYAB, LIM2, BHMT* and *PITX3* (Fig. 2j), well-documented contributors to lens development in vivo[25,27,28]. The mixed pool of glial and neural progenitors seen in WK2 SEAMs was likewise present at WK4. Neuronal cells in clusters 4, 5 and 11 expressed common elements such as *DCX, TUBB3* and *NEFM/L*, but also specific markers according to their developing subtype. For example, cells in cluster 4 expressed *SLC17A7* (*VGLUT1*) and *SLC17A6* (*VGLUT2*), which encode vesicular glutamate transporters and specifically mark glutamatergic neurons[46] while those in cluster 5 expressed the GABAergic markers *GAD1* and *GAD2*[47] (Fig. 2k), along with the DLX family members *DLX1, 2, 5* and *6*, which regulate GABAergic neuronal specification[48] (Supplementary Fig. 3). Cells in clusters 1 and 13 expressed markers indicative of populations of differentiating glial-like cells, including oligodendrocyte-like cells expressing *PLP1, FABP7* and *SOX3*, and astrocyte-like cells expressing *PAX2* and *VAX1* (Supplementary Fig. 3a, b). Surprisingly, only a small number of ocular epithelial cells were observed in our analysis of WK4 SEAMs, making up cluster 14 (Fig. 2h).

## Gene expression and cell lineage identity in advanced SEAMs; weeks 6–12

After culture for 6 weeks (WK6), differentiating iPSC cells are organised into a characteristic concentric multi-zone SEAM (Fig. 3a)[1]. Cells which reside in Zone 2 of 6-week-old SEAMs are known to represent those of the neural retina and RPE. Cluster analysis (Fig. 3b) revealed that like at WK4, there was a progressive delineation between RPE and neural retinal cells. Cells in cluster 5, and to a lesser extent, cluster 4, abundantly expressed RPE markers including *MITF, DCT, TYR* and *TYRP1* in a pattern which was increasingly non-overlapping with markers of neural retinal progenitor cells such as *VSX2, RAX* and *SOX2*, in clusters 0 and 1 (Fig. 3c). Clusters 7 and 8 represent proliferative progenitors (Supplementary Fig. 1). Retinal progenitor cells are multipotent, generating all six types of retinal neurons and Müller glia, in an ordered, sequential process, with retinal ganglion cells (RGCs) being the first to form. Cells in cluster 12 expressed *ATOH7* and *POU4F2*, which are required for RGC competence and differentiation[49], along with the neurotrophic receptor tyrosine kinase member *NTRK1* (Fig. 3d). Cells in cluster 11 expressed *ATOH7*, which marks RGCs in a transitional cell state of differentiation[50], and *VSX1*, which marks cone bipolar interneurons which connect photoreceptors to RGCs[51]. *ATOH7* also labels progenitors which give rise to early-born cone photoreceptors[52]. In human development, the birth of cone photoreceptors precedes that of rods, and cones can be identified by the expression of markers such as *PDE6H*, which labels a cone

cell-specific inhibitory subunit[53]. Cells in cluster 10 strongly and exclusively expressed *PDE6H*, along with the photoreceptor cell-type marker *CRX* (Fig. 3d), allowing us to identify spontaneously developing photoreceptors in iPSC-derived SEAMs. Relative to other clusters, cells in cluster 9 most strongly expressed *PTN, SPARCL1, LAMP5, VIM* and *PLP1* (Supplementary Fig. 4a, b). This glial-like transcriptional profile combined with robust expression of *PAX2, VAX1* (Fig. 3d) and *SOX2* led us to identify cells in this cluster as astrocyte precursor cells. Cells in clusters 3 and 6 strongly expressed neural markers in the absence of the optic markers *RAX, SIX6* and *VSX2*. While neurons in cluster 6 expressed the GABAergic markers *GAD1* and *GAD2* along with DLX-family transcription factors, those in cluster 3 abundantly expressed *HOX* genes, most notably of the *HOXB* family. Expression profiles for *HOXB* family members (Supplementary Fig. 4c) suggest that these cells are non-ocular and instead more representative of neurons of the hindbrain or anterior spinal cord.

Clusters 2, 13 and 14 at WK6 contain cells derived from the surface ectoderm. These expressed epithelial markers including *EPCAM, CDH1* and *TP63* (Fig. 3e). Cells in clusters 2 and 13 were transcriptionally similar, and expressed *KRT5*, a well-characterised marker of corneal epithelium, along with basal epithelial markers *GJB2* and *GJB6*[54,55] (Fig. 3f). Cells in both clusters also robustly expressed *PAX6* (Fig. 3e) and were therefore annotated as basal corneal epithelial cells. Cluster 13 represents mitotic cells of this subtype, characterised by *MKI67, TOP2A* and *UBE2C* expression (Supplementary Fig. 1). *CD200*, which is expressed during iPSC differentiation but absent from mature corneal epithelial cell lineages[56], was present at lower levels in epithelial clusters, and this pattern was maintained during the remainder of the SEAM culture protocol (Supplementary Fig. 5). Cells in cluster 14 expressed lower levels of *PAX6* but showed strong expression of *S100A9* and the keratins *KRT4, 7* and *13*, suggesting a non-ocular mucosal epithelial identity. This was corroborated by expression of the mucins *MUC4* and *MUC16* in these cells (Fig. 3f).

Cells from all four zones of the developing SEAM were well represented in our WK8 analyses (Fig. 3g, h). In Zone 2, expression in cluster 4 was dominated by the RPE marker *MITF* and factors known to be important for melanin production and RPE homeostasis[44], whereas *VSX2* expression was prevalent in the developing neural retina cells of cluster 0. (Fig. 3i). Cells in clusters 5 and 6 expressed mitotic markers (Supplementary Fig. 1a) together with *VSX2, PAX6* and *RAX* (Supplementary Fig. 6a) and were annotated as proliferative RPCs. Relative to other clusters, cells in cluster 8 most strongly expressed *SPP1, PLP1* and *PAX2* in combination with the RPC marker *VSX2* (Supplementary Fig. 6a). *SPP1* is expressed by immature Müller glia cells[57,58], which are the only glial cell type born from retinal progenitors, while *PAX2* has previously been shown to be expressed in chick, but not mammalian, Müller glia[59]. Meanwhile. *PLP1*, a commonly used label for enteric glia[60], has also been reported to be expressed by Müller glia[61]. It is possible, therefore, that immature Müller glia-like cells emerge at this timepoint in the SEAM. In common with WK6, there were clusters of astrocyte-like cells expressing *PAX2, VAX1* and *SOX2* (cluster 13), RGCs expressing *POU4F2, ATOH7, STMN2, NTRK1* and *NEFM* (cluster 12) and photoreceptors expressing *CRX, RCVRN, RXRG, PDE6H* and *NEUROD1* (cluster 11). *OTX2*, which is expressed by both photoreceptor cells and RPE and is essential for their development[62], was robustly expressed by cells in clusters 11 and 4 (Supplementary Fig. 6b, c). Clusters 3 and 9 contain *DCX, NEFL* and *NEFM*-expressing neuronal cells similar to those seen in younger

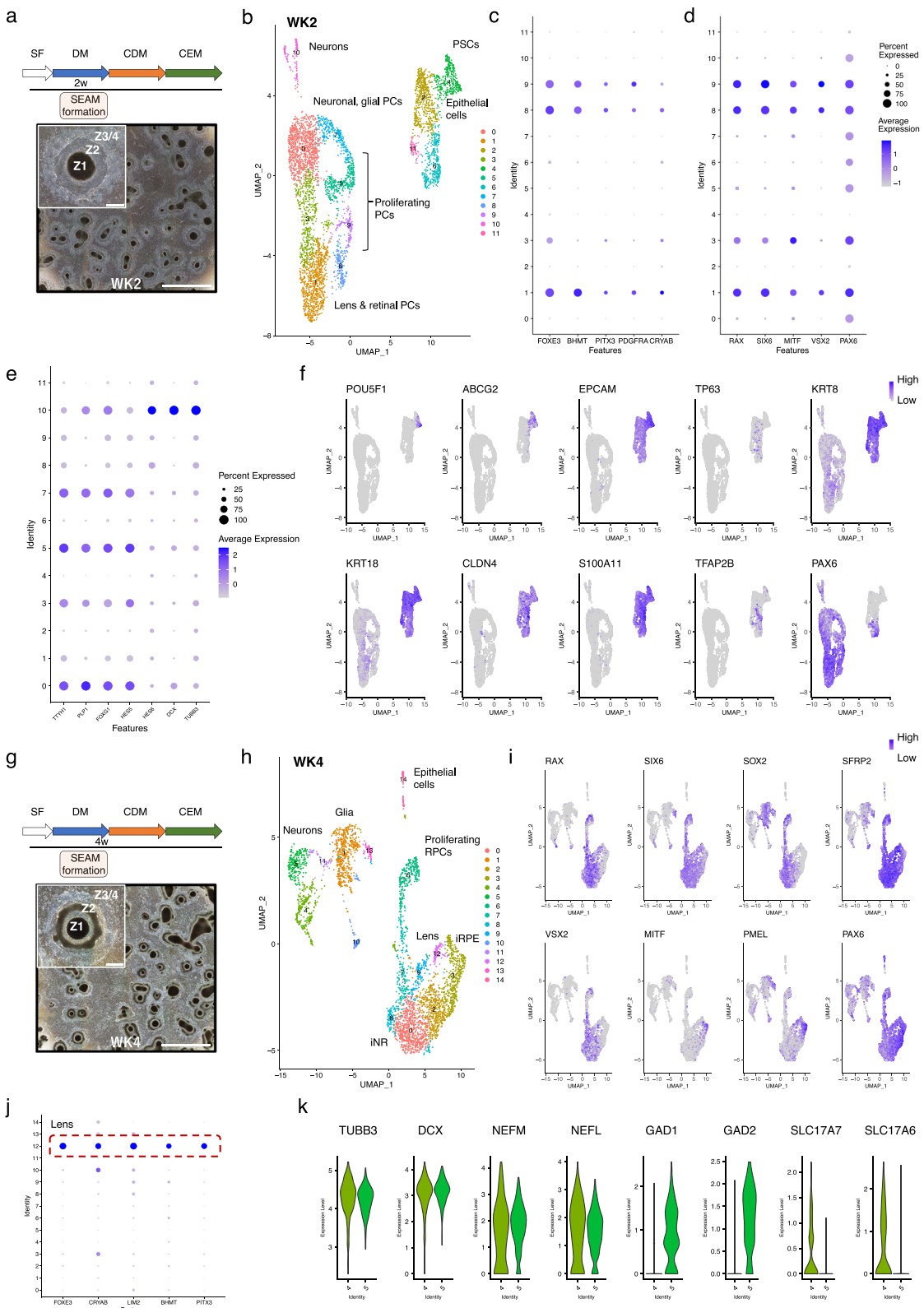

SEAMs, completing the Zone 1 profile. Compared with the neural retinal cells of Zone 2, cells in clusters 3 and 9 expressed much lower levels of *VSX2* and *RAX*, and are likely non-ocular in derivation. Meanwhile, *FOXG1* was expressed in both the neuronal (Zone 1) and retinal (Zone 2) clusters, reflecting its role in forebrain development and known expression in retinal tissues[63] (Supplementary Fig. 6b).

Clusters 2, 7 and 10 represent cells expressing the epithelial cell markers *CDH1*, *EPCAM* and *TP63*. Detailed analysis of these clusters revealed widespread expression of *KRT5* and *19*, with expression of the mucosal keratins *KRT4*, *7* and *13* restricted to a sub-population of cells (Fig. 3j). Further analysis of *PAX6* expression patterns suggests that cluster 10 contains a mixed population of cells, consisting of a small *PAX6*-positive

**Fig. 2 | scRNAseq analysis of WK2 and WK4 SEAMs. a** Experimental timeline and representative phase-contrast image showing WK2 SEAMs in culture. Scale bar: 5000 μm; inset, 500 μm. **b** UMAP plot of WK2 SEAMs. **c** Dot plots showing expression of lens progenitor cell markers *FOXE3*, *BHMT*, *PITX3*, *PDGFRA* and *CRYAB*. **d** Dot plots illustrating overlapping expression domains of NR markers and RPE marker *MITF*. **e** Dot plots showing expression of glial and neural markers. **f** Feature plots showing populations of stem (*POU5F1*, *ABCG2*) and surface ectoderm-derived epithelial cells. Elevated *PAX6* expression indicates ocular lineages. **g** Experimental timeline and representative phase-contrast image showing

WK4 SEAMs in culture. Scale bar: 5000 μm; inset, 500 μm. **h** UMAP plots of WK4 SEAMs. **i** Feature plots showing widespread expression of *PAX6, SIX6 and SFRP2* and reciprocal expression of *VSX2, SOX2* and *RAX* in developing NR compared with *MITF* and *PMEL* in RPE. **j** Lens marker expression in Seurat cluster 12. **k** Violin plots showing contribution of neuronal cell subtypes to clusters 4 and 5. Z1 Zone 1, Z2 Zone 2, Z3 Zone 3, Z4 Zone 4, SF StemFit medium, DM differentiation medium, CDM corneal differentiation medium, CEM corneal epithelium maintenance medium, PSC pluripotent stem cell, PC progenitor cell, RPC retinal progenitor cell, NR neural retina, RPE retinal pigment epithelium. 'i' denotes immature cells.

population of cells which co-express *KRT13* and likely represent con-junctival epithelium[64], alongside a larger proportion of Zone 4 *PAX6*-negative cells belonging to non-ocular mucosa. Cells in clusters 2 and 7 were transcriptionally similar and robustly expressed *PAX6* and *KRT5* in the absence of *KRT13*, indicating a Zone 3 corneal epithelial phenotype[55,64], with substantial expression of the basal corneal epithelial cell markers *GJB2* and *GJB6* restricted to cells in cluster 2 (Fig. 3j). The cells in the smallest cluster at this stage, cluster 14, represent a population of endothelial-like cells. These cells expressed *PAX6* and high levels of *COL3A1*, *TM4SF1*, *SPRR2F*, *GATA6* and *PODXL* in addition to *COL8A1*[65], a corneal endothelial marker. Expression of *NPNT*, which has been reported to play a role in the migration of periocular neural crest cells during chick corneal development[66], and *MGP*, which is expressed in both the trabecular meshwork and the corneal endothelium[67], was also observed in these cells (Fig. 3k).

In WK10 SEAMs (Fig. 4a, b) there was further demarcation between the NR and RPE cell clusters, with expression of RPE markers restricted to cluster 5 (Fig. 4c). Photoreceptors expressing *CRX*, *RCVRN*, *RXRG*, *THRB* and *PDE6H* make up cluster 14, whilst RGCs expressing *POU4F2*, *ATOH7* and *NTRK1* are found in cluster 10. *PAX2 + /VAX1+* astrocytic cells are located in cluster 6, and these cells co-expressed *SOX2* and glial-like markers such as *PLP1*, *LGI4*[68], *SPARCL1* and *METRN*[69,70]. Similar to in WK8 SEAMs, a small population of cells (cluster 15) expressed the RPC marker *VSX2* together with glial cell-type markers including *PAX2* and *PLP1*, and therefore likely represent early Müller glia progenitors (Fig. 4c). At this developmental timepoint, clusters 7 and 13 are comprised of proliferative retinal progenitor cells co-expressing *VSX2*, *SOX2*, *RAX* and *SFRP2* in combination with mitotic markers (Supplementary Fig. 1). A small number of *HOX* gene expressing neurons co-expressing prototypical markers including *TUBB3*, *DCX* and *NEFL/M* make up cluster 16 (Fig. 4c).

Epithelial cell populations derived from surface ectoderm were well represented at WK10, with 2045/4260 (48%) of retained cells belonging to this lineage. Six clusters of *EPCAM* and *CDH1*-expressing epithelial cells were present, with various keratin expression profiles illustrated in Fig. 4d. Cells in cluster 1 strongly expressed *PAX6*, *TP63*, *KRT5* and the basal epithelial markers *GJB2* and *GJB6* (Fig. 4d, e) and were annotated as basal corneal epithelial cells. Cells in cluster 8 were *PAX6+ /KRT5+* and were transcriptionally similar to those in cluster 1. However, these cells also expressed markers typically associated with cell migration and corneal wound healing, including *MMP10*[71] and *POSTN*[72] (Fig. 4e). It is conceivable that this transcriptional profile is applicable to cells undergoing tissue remodelling or migration during development. Cells in cluster 11 expressed *PAX6*, the mucosal chemokine *CXCL17*[73] and the functional lacrimal gland markers *LCN2* and *DEFB1*[6]. The remaining clusters (4, 9, 12) contain cells typically found in Zone 4 of the SEAM. These cells expressed lower levels of *PAX6* and are likely non-ocular in derivation. Cells in clusters 4 and 9 expressed *DSG3*, which is found in squamous epithelium in skin and oral mucosa[74]. Some cells in cluster 9 expressed *KRT1* and increased levels of its heterodimer partner *KRT10*, both of which are expressed in suprabasal keratinocytes in the skin[75], while expression of the mucociliary epithelial marker *AGR3*[76], was observed in cells in cluster 12 (Fig. 4e).

In the WK12 SEAM (Fig. 4f, g), the sequential emergence of retinal neurons continued. Clusters 0, 3 and 6, for example, contain retinal progenitor cells at varying stages of maturation. Cells in cluster 7 expressed various RPE markers including *RLBP1*, *BEST1*, *MITF*, *MLANA*, *SFRP5*, *COL8A1*, *WNT2B* and *TYR*. However, we were unable to reliably detect

expression of the late RPE marker *RPE65*. (Fig. 4h, Supplementary Fig. 7). Cells in cluster 14 strongly expressed multiple photoreceptor markers and high levels of *TLCD3B*, a gene recently discovered to be associated with recessive retinal dystrophy[77]. RGCs in cluster 10 expressed the prototypical markers *POU4F2*, *ATOH7*, *SLC17A6*, *STMN2* and *NTRK1* together with the homeobox transcription factors *ONECUT1* and *ONECUT2*, which function during development in the specification of RGCs and horizontal cells[78] (Fig. 4h). Cluster 13 represents *PAX2 + /VAX1 + /SOX2+* astro-cytes, while cells in cluster 15 expressed glial-like markers including *SPARCL1*, *METRN* and *PTN*, together with posterior *HOX* family members such as *HOXB8* and *HOXB9* (Fig. 4h).

Clusters 1 and 11 represent *PAX6*-positive corneal epithelium robustly expressing *KRT5* but not *KRT 4, 7* or *13*. While cells in both clusters expressed *TP63*, *GJB2* and *GJB6*, indicating a basal origin, cells in cluster 11 also expressed *KRT12*, which marks terminally differentiated corneal epi-thelial cells[79], indicating a more fully developed phenotype (Fig. 4i, j). Cells in cluster 9 were *KRT7 + /KRT13+* and expressed *CXCL17* together with the mucosal markers *MUC4* and *MUC16* and the lacrimal gland markers *DEFB1* and *LCN2* (Fig. 4j), suggesting that these are conjunctival epithelial and lacrimal gland cells[6,80]. Cells in cluster 12 expressed mitotic markers (Supplementary Fig. 1) along with *PAX6* and *KRT5*, and are presumably proliferating corneal epithelial cells, whilst those in cluster 8 had a tran-scriptional profile similar to the migratory cell population described in WK10 SEAMs. The putatively annotated non-ocular clusters of stratified oral mucosal and skin epithelial cells were also present at WK12, located in clusters 2 and 4. Heatmaps showing the top 5 genes expressed by cells of each of the returned clusters at each timepoint are shown in Supplementary Fig. 8 and all results generated by FindAllMarkers are available in Supple-mentary Data 1.

## Integrative analysis of developmental trajectories

In order to follow changes in gene expression as differentiation within the SEAM progressed, individual Seurat objects were merged and normalised to correct for heterogeneity in sequencing depth. For clarity, and due to the over-representation of neuronal cells at WK4, this timepoint was excluded from the combined analyses. UMAP representations of the combined dataset were prepared according to Seurat cluster (Fig. 5a) and original timepoint identity (Supplementary Fig. 9a), and differential expression analysis was performed to reveal specific cluster markers (Supplementary Data 2). We next used Monocle3 to identify modules of co-regulated genes which may drive differentiation (Fig. 5b, Supplementary Data 3) and to explore cellular dynamics by performing trajectory and pseudotime inference[81]. Figure 5c shows a clear trajectory progression along two prin-cipal pathways which emerge as cells transition from a pluripotent state at WK0 into either a surface ectodermal or neuroectodermal lineage, even-tually giving rise to the multiple cell types depicted in Fig. 5a. These cell clusters are shown ordered according to Monocle3 pseudotime in Fig. 5d, beginning with pluripotent cells in cluster 10 and ending with photo-receptors in cluster 21. Pseudotime values were assigned to all cells in the UMAP space (Fig. 5e), and trajectory-variable genes were collated into co-expression modules (Supplementary Fig. 9b and Supplementary Data 4). These analyses provide a reconstruction of linked developmental trajec-tories from SEAM transcriptomic data and additionally create a resource to identify potentially novel drivers of module and cluster-specific cellular differentiation which may underpin mechanisms of eye development.

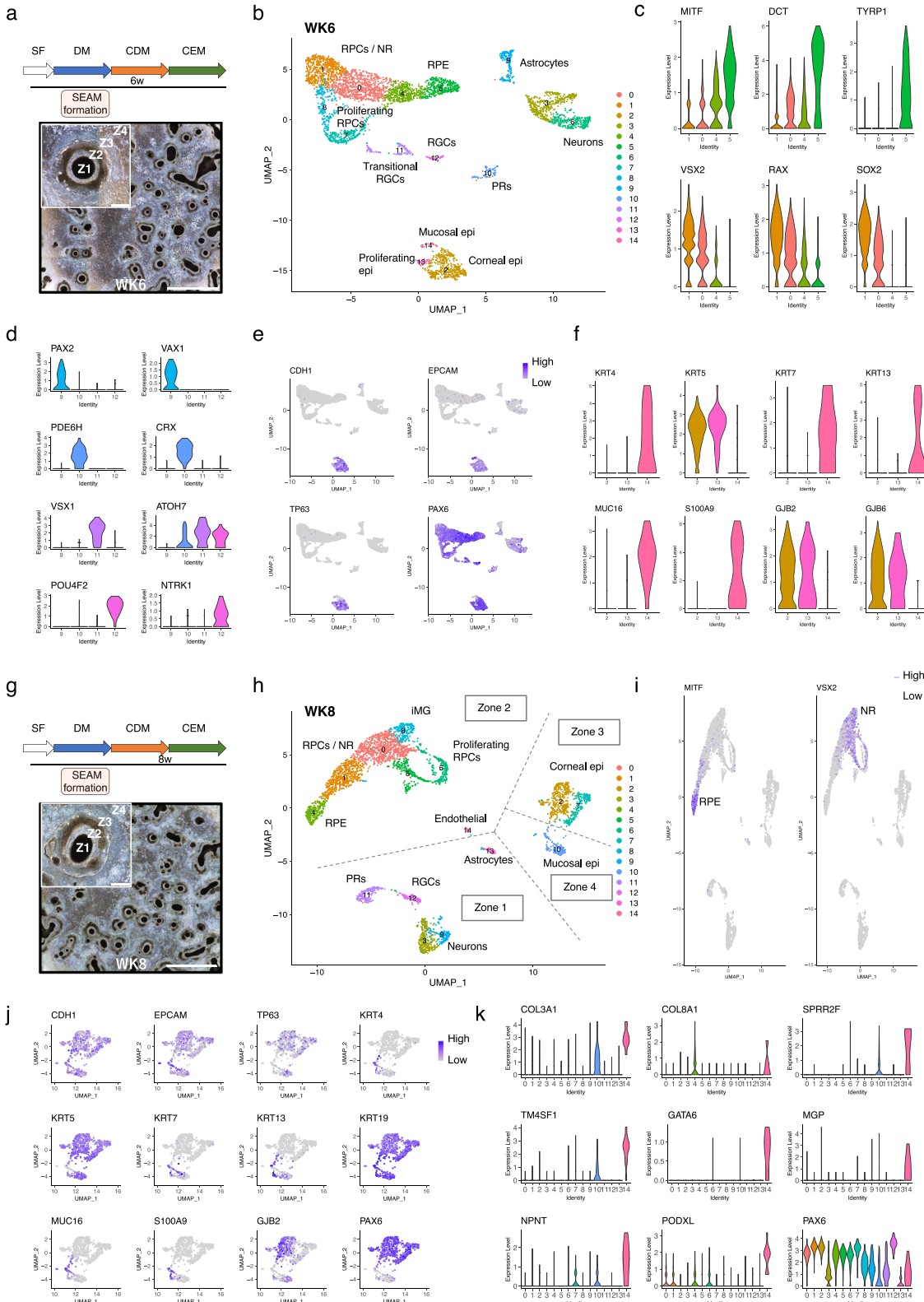

## Discussion

Here, we provide a comprehensive prolife of the transcriptomic landscape of hiPSC-derived SEAMs as they differentiate and grow in vitro as representations of human eye development. We have identified and molecularly characterised cellular populations which form within each of the concentric zones of the growing SEAM and have observed the generation of highly specialised cell types such as RGCs and photoreceptors, highlighting the intrinsic potential of this system to spontaneously generate multiple ocular tissues. Our data also reveal that two lineages emerge soon after the cells transition from a pluripotent state. Indeed, even at the initial WK0 timepoint, presumptive neural and epithelial lineage cells can be distinguished, likely because the hiPSCs have begun to partially differentiate during their

**Fig. 3 | scRNAseq analysis of WK6 and WK8 SEAMs. a** Experimental timeline and representative phase-contrast image showing WK6 SEAMs in culture. Scale bar: 5000 μm; inset, 500 μm. **b** UMAP plot of WK6 SEAMs. **c** Violin plots illustrating progressive delineation of developing RPE (*MITF, DCT, TYRP1*) and NR (*VSX2, RAX, SOX2*). **d** Violin plots showing expression patterns indicative of astrocytes (*PAX2, VAX1*), RGCs (*POU4F2, NTRK1, ATOH7*), bipolar interneurons (*VSX1*) and cone photoreceptors (*PDE6H, CRX*). **e** Feature plots showing expression of epithelial markers. **f** Violin plots showing expression of corneal (*KRT5*), basal epithelial (*GJB2, GJB6*) and mucosal (*KRT4, KRT7, KRT13, MUC16, S100A9*) markers in epithelial subsets. **g** Experimental timeline and representative phase-contrast

image showing WK8 SEAMs in culture. Scale bar: 5000 μm; inset, 500 μm. **h** UMAP plot of WK8 SEAMs indicating contribution of clusters to SEAM zones. **i** Feature plots showing non-overlapping domains of NR and RPE cells in the SEAM. **j** Feature plots showing expression patterns in subsets of epithelial cells. **k** Violin plots showing expression of endothelial markers in cluster 14. Z1 Zone 1, Z2 Zone 2, Z3 Zone 3, Z4 Zone 4, SF StemFit medium, DM differentiation medium, CDM corneal differentiation medium, CEM corneal epithelium maintenance medium, PSC pluripotent stem cell, epi epithelium, PC progenitor cell, RPC retinal progenitor cell, NR neural retina, RPE retinal pigment epithelium, RGC retinal ganglion cells, PR photoreceptors, MG Müller glia. 'i' denotes immature cells.

initial 10-day pre-cultivation in StemFit medium. By WK1 of SEAM formation, distinct partitions representing either presumptive surface ectoderm or neuroectoderm are present. The predominance of cells with a neuroectodermal signature at this stage is in accordance with previous immunohistochemical findings showing that the innermost zone of the SEAM, the neuroectodermal Zone 1, is the first to form[1]. The current data indicate an underlying genomic bifurcation (i.e. surface ectoderm vs. neuroectoderm) that signifies an early commitment to distinct cellular lineages.

By WK2 of SEAM formation there is significant overlap in clusters containing progenitors of both lens and retinal cells. In vivo, the optic vesicle (which gives rise to the neural retina and RPE) and the surface ectoderm (which forms the lens) are initially in close proximity and are influenced by similar sets of signals. It is likely that as development progresses and the tissue architecture begins to establish, these progenitor cells receive more specific cues leading to their spatial and functional delineation, and this is perhaps similarly occurring during SEAM development. In support of this notion is the initial overlap in cells expressing the RPE marker *MITF* and the retinal progenitor cell marker *VSX2*. It is well established that *MITF* is initially expressed uniformly throughout the optic vesicle and that repression by VSX2 is a critical step in delineation of RPE and neuronal retinal territories[32], and this is strikingly well conserved during SEAM development and maturation. Mixed progenitor pools of glia and neurons are also evident at this stage. During embryonic neurogenesis, pools of multipotent progenitor cells give rise to both neurons and glial cells in a carefully coordinated manner. The intricate balance between proliferation and differentiation of these progenitors is crucial for the generation of the appropriate numbers of neurons and glia, suggesting that these clusters represent a transitional state of progenitor cells within the SEAM that are likewise actively engaged in the process of fate determination.

The large proportion of cells belonging to a neural or glial identify at WK4 was initially surprising, particularly as this was accompanied by an unexpectedly small ocular epithelial cell population. However, it should be noted that during the SEAM differentiation protocol, it is at this point (i.e. WK4) that the initial differentiation medium (DM) is substituted for corneal differentiation medium (CDM)[2], which contains KGF and the ROCK inhibitor Y-27632. KGF is a well characterised mitogen which specifically simulates the growth and survival of epithelial cells[82], while Y-27632 has been reported to promote the proliferation of limbal epithelial cells and acceleration of corneal wound healing[83]. It is conceivable, therefore, that the results obtained at WK4 reflect SEAM cultures before the addition of factors which are critical for their sustained epithelial differentiation. For this reason, and in the interest of clarity, we excluded the WK4 data from our trajectory analyses. By WK6, a stage at which growing SEAMs are easily identifiable by their characteristic concentric multi-zones[1,2], epithelial cell populations are well represented. Furthermore, it is possible to identify distinct populations of corneal and mucosal epithelium, and it is likely that the growth and survival of epithelial cells from this point in the differentiation protocol is reliant on the change in media to CDM, and later, to corneal epithelial maintenance medium (CEM). This is supported by the subsequent expansion of these epithelial populations and appearance of cell clusters expressing markers of functional lacrimal gland at later stages.

*SOX10* + /*p75*+ neural crest cells have previously been identified in satellite spheres in immunohistochemical analyses[1] and our analysis has revealed evidence of neural crest-derived periocular mesenchyme in

immature SEAMs, consistent with this finding. Corneal endothelial cells derive from this periocular mesenchyme, and we have identified endothelial-like cells in very limited numbers in WK8 SEAMs. However, these cells are not maintained, again indicating that additional modifications to the protocol might be needed in order to support their generation and continued development. The corneal stroma is also derived from periocular mesenchyme, and although we observed widespread expression of the stromal marker *DCN*, there was no significant expression of the classical stromal markers *KERA* and *LUM*. Given that *DCN* has also been reported to be expressed in the limbus and peripheral cornea and in the skin[84], it is unlikely that these cells represent stromal keratocytes, but we cannot exclude the possibility that a small population of stromal cells are found interspersed within the epithelial clusters.

As the SEAM matures, discrete clusters representing specialised cell types such as RGCs or photoreceptors become readily identifiable due to their distinct transcriptomic signatures. Furthermore, temporal specification appears to be largely conserved during SEAM growth and maturation. For example, RGCs, developmentally the earliest born of the retinal neurons, first emerge as a distinct population in WK6 SEAMs. Retinal neuron subtypes are generated in chronologically overlapping waves, and cones, which are born shortly after RGCs, are also identifiable in the SEAM at this stage. Some cell populations in our dataset are more challenging to classify. For instance, we observed a population of cells co-expressing retinal progenitor markers like *VSX2* and *RAX* along with markers indicating a glial-like identity. These cells likely represent immature Müller glia, considering their developmental origin from retinal progenitor cells. However, it is notable that these cells express *PAX2*, which is not typically associated with mammalian Müller glia[59], while classical Müller glia markers such as *GLUL*, *CLU* and *RLBP1* were either absent or showed no specific upregulation. Overlapping expression profiles between developing Müller glia and retinal progenitor cells are well-documented[85], and transcriptomic studies of hESC-derived retinal organoids indicate early emergence of Müller glia-like cell populations during retinal organoid development[86]. Evaluating the upregulated genes within this cluster may shed light on the mechanisms driving functional differentiation of these immature cells. We also noted the absence of some other mature cell-type-specific markers in our data. For example, in WK12 SEAMs, while we were able to detect expression of multiple prototypical RPE markers, we were unable to reliably detect expression of *RPE65*, which typically marks 'late' RPE[87]. Our SEAM culture protocol drives cell autonomous ocular differentiation in the absence of exogenous stimuli, but mature RPE has successfully been generated from hiPSCs by using defined media components and by using modified protocols which promote retinal organoid formation and expansion of the RPE[88–90]. Similarly, while we were able to follow the emergence and continued development of photoreceptors in our cultures, we were unable to detect expression of rhodopsin (*RHO*), or the cone opsins (*OPN1SW, OPN1MW* and *OPN1LW*), suggesting that these cells may not have reached maturity or that the culture conditions are suboptimal for promoting expression of these specific genes at the timepoints studied. Indeed, while opsin-expressing photoreceptors have been generated from hiPSCs using specific protocols, detectable levels of these markers typically emerged only after the period covered by our study[91–93]. For example, it has been reported that photoreceptors derived from hiPSCs under directed pro-neural differentiation methods do not express mature markers until week 14 of culture, while in hiPSC-derived optic cups, opsin

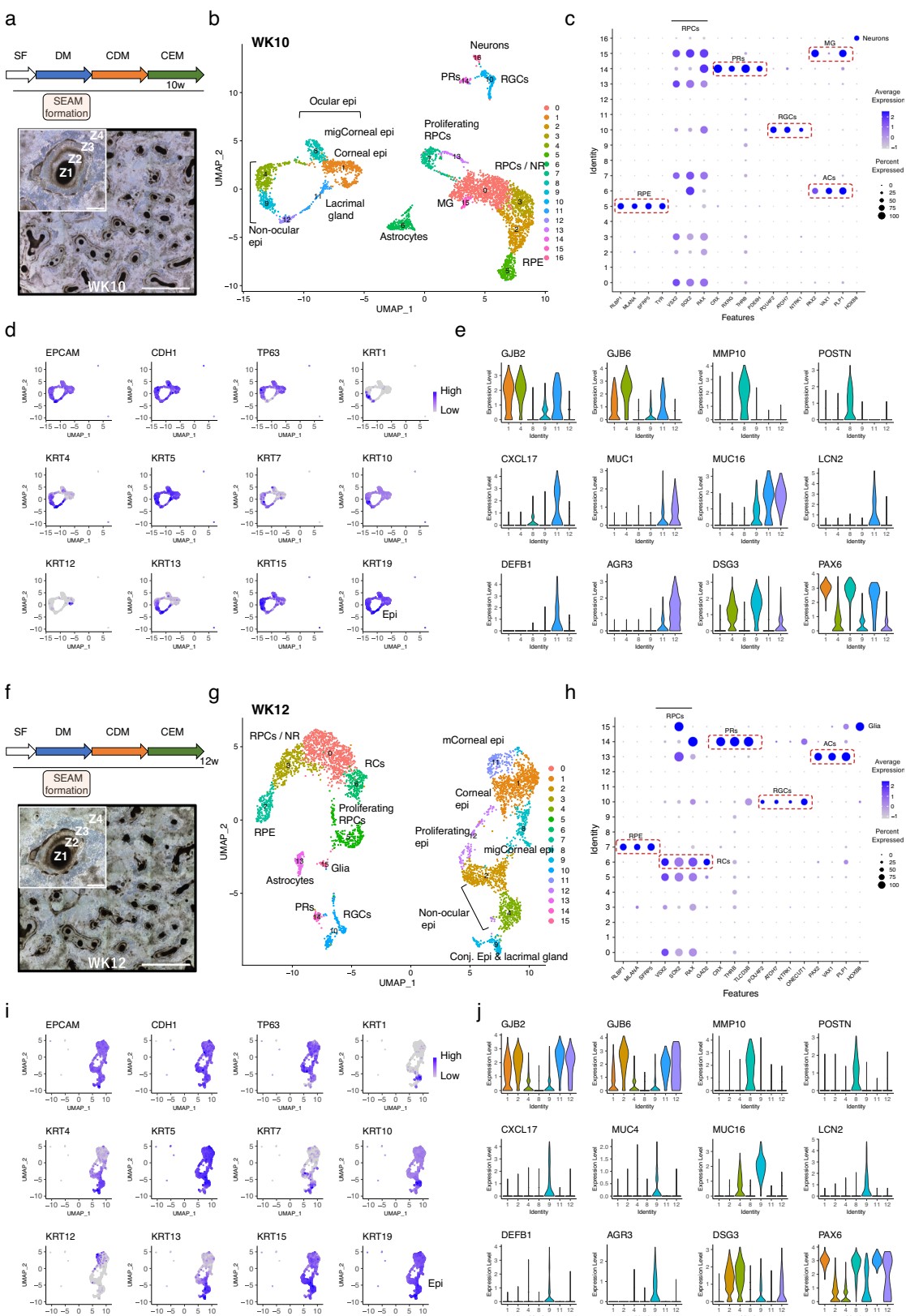

expression is not detected until week 21[94,95]. The absence of opsin expression in our data, therefore, could reflect intrinsic developmental timing or be because our two-dimensional adherent cell culture protocol and standard media composition does not support expression of a full complement of mature photoreceptor markers without the inclusion of growth factors to promote continued differentiation.

The UMAP projection during SEAM retinal cell maturation reveals a distinct transition from clusters characterised by RPE markers to those of a neural retinal identity. Notably, cells clustered around the border of this transition frequently expressed genes such as *GJA1*, *OTX1*, *AQP1* and *ZIC1*, suggestive of a ciliary margin identity. Ciliary margin-like niches have previously been reported to form in hESC and hiPSC / SEAM culture

**Fig. 4 | scRNAseq analysis of WK10 and WK12 SEAMs. a** Experimental timeline and representative phase-contrast image showing WK10 SEAMs in culture. Scale bar: 5000 μm; inset, 500 μm. **b** UMAP plot of WK10 SEAMs. **c** Dot plot showing expression profiles of specialised cells in the SEAM. **d** Feature plots showing expression of epithelial markers and keratin subtypes across clusters. **e** Violin plots illustrating expression of basal epithelium (*GJB2, GJB6*), lacrimal gland (*LCN2, DEFB1*), skin & mucosal (*DSG3, CXCL17, MUC1, MUC16*), mucociliary (*AGR3*) and migratory cell (*MMP10, POSTN*) markers. **f** Experimental timeline and representative phase-contrast image showing WK12 SEAMs in culture. Scale bar: 5000 μm; inset, 500 μm. **g** UMAP plot of WK12 SEAMs. **h** Dot plot showing

expression profiles of specialised cells in the SEAM. **i** Feature plots showing expression of epithelial markers and keratin subtypes across clusters. **j** Violin plots illustrating expression of basal epithelium (*GJB2, GJB6*), lacrimal gland (*LCN2, DEFB1*), skin & mucosal (*DSG3, CXCL17, MUC4, MUC16*), and migratory cell (*MMP10, POSTN*) markers. Z1 Zone 1, Z2 Zone 2, Z3 Zone 3, Z4 Zone 4, SF StemFit medium, DM differentiation medium, CDM corneal differentiation medium, CEM corneal epithelium maintenance medium, epi epithelium, mig migratory, PC progenitor cell, RPC retinal progenitor cell, NR neural retina, RPE retinal pigment epithelium, RGC retinal ganglion cells, PR photoreceptors, ACs astrocytes, RCs retinal cells, MG Müller glia. 'm' denotes mature cells.

systems using modified protocols[88,96] but our results imply that this region has the potential to spontaneously generate independently without culture modifications. The ciliary marginal zone as a source of retinal stem and progenitor cells has been well studied in lower vertebrates such as fish, frogs and birds, although the functional presence of an analogous structure in the mammalian eye is broadly debated[97]. Nevertheless, this remains an interesting focus for future study and once again highlights the spontaneous co-induction of different ocular regions in the absence of exogenous factors.

In previous work, Zone 4 of the SEAM has been identified as non-ocular surface ectoderm, owing to expression of epithelial markers such as *CDH1* and *TP63* in the absence of *PAX6*. It was proposed that these cells would likely differentiate into epidermal keratinocytes[1,5]. P63+ cells are able to terminally differentiate into stratified epithelium of the cornea, skin, or oral mucosa[20,41], with an ocular phenotype being dependent on the expression of *PAX6*. Oral mucosal epithelium and ocular surface epithelium share many phenotypic and morphological characteristics, and oral mucosal epithelial cell sheets have been used to reconstruct damaged ocular surface in limbal stem cell deficiency models and have shown good efficacy in clinical trials[98–100]. Our analysis has also revealed transcriptional similarity between clusters of ocular epithelial and putative oral mucosa cells and suggests that these clusters may contain a mixed population of cells. This is illustrated by the relatively low overall expression of *PAX6* in the annotated conjunctival epithelial cell cluster, although it should be noted that lower levels of endogenous *PAX6* expression in the conjunctiva compared with cornea have also been previously reported[98]. Mixed populations of conjunctival and non-ocular epithelium have additionally been identified in the same fraction following FACS analysis to isolate specific ocular surface lineages from growing SEAMs, and HOXB4 expression was seen in cells which did not express PAX6[1]. This is mirrored in our recent data which shows expression of several *HOX* genes, which are not expressed at the ocular surface, in isolated cells belonging to epithelial clusters. We have also found expression of *HOX* genes in other groups of cells, most notably in neuronal cell clusters. *HOX* genes are not expressed in the retina[101], so these *HOX*-expressing neuronal cells represent a broader neuroectodermal differentiation capacity within the SEAM. The *HOX*-expressing neuronal cluster at WK6, for example, shows robust and widespread expression of *HOXB* members *HOXB2* through to *HOXB8*. *HOXB2* is expressed in developing hindbrain rhombomeres and plays a role in patterning during neurogenesis[102], while *HOXB8* is expressed more posteriorly in the dorsal spinal cord[103], indicating that some cells may be defaulting to a more posterior neuroectodermal developmental pathway, possibly in response to intrinsic or extrinsic cues that are yet to be defined. One potential candidate for this is retinoic acid (RA). *HOX* genes are long established to be direct effectors of RA signalling in embryogenesis and development, and RA signalling confers posterior identity during patterning of the developing nervous system[104]. RA also plays a number of roles in vertebrate eye development[105], and consistent with this, our data show that the RA-metabolising enzymes *RDH10* and *ALDH1A1 – ALDH1A3* are expressed in developing SEAMs.

Analysis of time-series transcriptomic data from differentiating stem cells is a powerful approach to further our understanding of the dynamic and temporal changes in gene expression which shape developmental processes[106–108]. Our results here illustrate progressive development of SEAMs from iPSCs over a 12-week period and provide detailed analyses at

key junctures which follow the transition from initial pluripotency though to generation of specialised ocular tissues. Further, we provide a clear snapshot of the contribution of cells from SEAMs of different ages and zones to the range of structures generated. By reconstructing developmental trajectories and extracting modules of co-regulated genes we are also able to identify less well studied candidate genes, highlighting the usefulness of these tools in examining potential drivers of specific differentiation pathways and offering important avenues for further study. While this work has allowed for us to explore, in fine molecular detail, the formation of the SEAM in vitro and the establishment of its characteristic zones, it has also revealed remarkable complexity and the innate self-directed differentiation of ocular components along divergent lineages at a transcriptional level. Further, our study not only advances our understanding of ocular development but also establishes a robust methodological framework for examining and customising SEAM development in vitro.

## Methods
### SEAM cell culture
SEAMs were generated from human iPS cells (clone 201B7[7], RIKEN BioResource Center, Tsukuba, Japan) as described previously[1,2,6]. Briefly, hiPSCs were maintained on 0.5 μg/cm² LN511E8 (iMatrix-511 silk, 892021, Nippi, Tokyo, Japan) coated culture dishes in serum-free StemFit medium (AK03N, Ajinomoto, Tokyo, Japan) for at least 2 cell passages for stabilisation. Cells were harvested using dissociation solution (DS) containing 50% TrypLE Select (13567-84, Nacalai Tesque, Kyoto, Japan) and 50% 0.5 mM EDTA/PBS solution (13567-84, Nacalai Tesque) and seeded at 4500 cells per well in 6 well culture plates (353046, Corning, NY, USA) coated with 0.5 μg/cm² LN511E8. Cells were then cultivated for a further 10 days in StemFit medium, and differentiation was initiated by culture in differentiation medium (DM) following established protocols[2]. After culture for 4 weeks, DM was substituted for corneal DM (CDM) containing growth factors (KGF, 112-00813, Wako Pure Chemical Corporation, Osaka, Japan) and the ROCK inhibitor Y-27632 (030-24-26, FUJIFILM Wako Pure Chemical Corporation) to promote epithelial cell growth and survival. After an additional 4 weeks of CDM culture (8 weeks in total), the medium was changed to corneal epithelial maintenance medium (CEM) consisting of DMEM/F12 (1:1; 11320033, Life Technologies, Carlsbad, CA, USA) containing KGF, Y-27632 and B-27 supplement (17504-044, Life Technologies). Medium changes were performed once every 2-3 days throughout the SEAM cell culture period. The composition of the culture medium for each stage of SEAM culture is shown in Supplementary Table 1.

### Single-cell library generation and sequencing
Whole SEAMs were harvested from individual wells at eight successive intervals spanning 0 – 12 weeks (every 2 weeks, with the addition of the WK1 sample to capture very early differentiation) in the same single series experiment using methods optimised for cells at each timepoint. Specifically, cells were harvested (1 well/timepoint) using dissociation solution (DS; WK0, 4 min), TrypLE Express (13567-84, Nacalai Tesque; WK1, 10 min) or Accutase (12679-54, Nacalai Tesque; WK2, 10 min; WK4 – WK12, 60 min). For every timepoint, harvested single-cell suspensions containing cells from all zones were sorted using a FACSAriaII cytometer (BD Biosciences, Franklin Lakes, NJ, USA) and living cells were selected by 7'AAD (559925, BD Biosciences) staining. Sorted single-cell suspensions

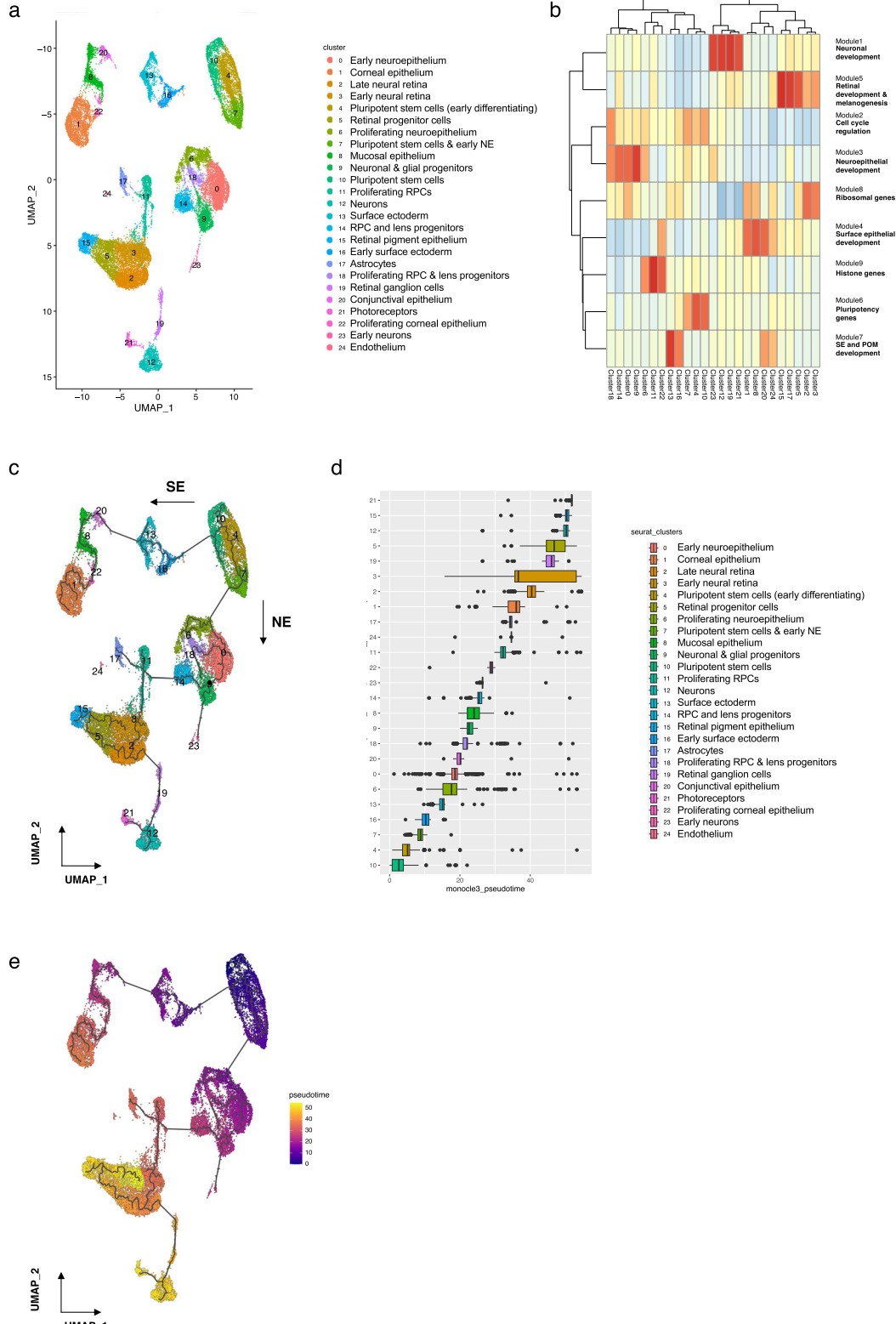

**Fig. 5 | Combined data and trajectory inference. a** UMAP representation of combined data, grouped by Seurat cluster. **b** Clustered heatmap showing aggregated expression of all genes in Monocle3 modules across Seurat clusters. Co-expressed genes were clustered into modules using find_gene_modules and heatmaps were generated using the 'pheatmap' package. **c** Monocle3 trajectory inference predicted by learn_graph. Surface ectodermal vs. neuroectodermal lineages are indicated by arrows. **d** Boxplot showing distribution of Monocle3 pseudotime values within each Seurat cluster, with clusters reordered based on their median pseudotime values. **e** UMAP plot with cells coloured by pseudotime. The root of the trajectory is labelled (1). SE surface ectoderm, NE neuroectoderm.

were prepared according to current 10x Genomics guidelines and single-cell libraries were prepared using Chromium Next GEM Single Cell 3′ Reagent Kits with v3.1 chemistry (16 rxns PN-1000127, 10x Genomics, Pleasanton, CA, USA) following the manufacturer's protocols. Single cells were partitioned into GEMs in a Chromium Single Cell Controller at a recommended concentration for targeted recovery of 5000 cells per timepoint, and libraries were constructed according to published user guidelines. Following library construction, libraries were sequenced with 10x Genomics dual indexing on an Illumina NovaSeq 6000 platform using a high-output flow cell to obtain paired-end reads.

## Single-cell data processing and analysis

Sequencing data files were transformed into single-cell gene count matrices with Cell Ranger 6.1.1 using default parameters and mRNA reads were mapped to the human reference genome GRCh38-2020-A (Supplementary Fig. 10a, b). Data pre-processing and cell cluster analyses were performed in R using Seurat v4[109,110]. Briefly, quality control metrics were first applied to the raw data to filter out cells with unique feature count values less than 500 or with greater than 15% mitochondrial reads (Supplementary Fig. 10c). Any additional putative low-quality cells were filtered out post-processing and heterotypic doublets were excluded using DoubletFinder[111] with parameters: PCs (principal components) = 1:50, pN (defined doublet proportion) = 0.25, pK (PC neighbourhood size) = pK_choose. pK was adjusted programmatically for each dataset to provide optimal values. Data were normalised and scaled using SCTransform v2[8] and dimensionality reduction was performed by PCA and UMAP embedding. Clustree[112] was used to assess cluster stability and a clustering resolution of 0.8 with 50 PCs was used in all downstream analyses. Differential expression analysis was performed using the FindAllMarkers function with min.pct = 0.25 and logfc.threshold = 0.25. Clusters were annotated according to differential expression profiles with reference to canonical markers, and where indicated, were correlated with respective SEAM zones by comparison with known SEAM zone markers and cell phenotypes. For the combined analysis, individual Seurat objects were integrated using the merge function, and PrepSCTFindMarkers was applied before differential expression analysis in order to correct for heterogeneity in sequencing depth. Monocle3[81,113] was used for trajectory and pseudotime inference. Briefly, the as.cell_data_set function from SeuratWrappers was used to convert a Seurat object into a Monocle data structure and learn_graph() was used to fit a principal graph. The root of the trajectory was determined programmatically and order_cells() was applied to order the cells in pseudotime. Differential expression analysis was performed using graph_test() with neighbor_graph = "knn". Modules of co-regulated genes were extracted using find_gene_modules with resolution=1e-4. To analyse genes which change as a function of pseudotime, graph_test() was passed to neighbor_graph = "principal_graph" and find_gene_modules() applied with resolution=1e-4.

## Statistics and reproducibility

Statistical analyses were performed using R version 4.1.2 (The R Project for Statistical Computing) and designated packages. Data were obtained from single cells with targeted recovery of 5000 cells per time point in a single-series experiment. To ensure reproducibility, SEAMs were cultured using established protocols and closely monitored for the appearance of concentric multi-zones. Single-cell expression data confirmed the consistent generation of ocular cell types over continuous cellular states across all samples.

## Reporting summary

Further information on research design is available in the Nature Portfolio Reporting Summary linked to this article.

## Data availability

scRNA-seq datasets have been deposited in NCBI GEO under accession number GSE263987. Source Data are provided in Supplementary Data files 1–4.

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

## Acknowledgements
This work was supported by the BBSRC (BB/S015981/1, BB/R021244/1) with contributions from Grant-in-Aid for Scientific Research (20H03842, 23H03060), Japan Society for the Promotion of Science (JSPS) and Fusion Oriented Research for Disruptive Science and Technology (JPMJFR210W), Japan Science and Technology Agency (JST). For the purpose of open access, the author has applied a CC BY public copyright licence to any Author Accepted Manuscript version arising from this submission.

## Author contributions
M.J.H., D.J.B., K.N., R.H. and A.J.Q. conceptualised the research. Y.I. and T.K. performed the in vitro SEAM experiments and acquired the data. L.H. analysed the scRNA-seq data with advice from S.-J.P. and wrote the manuscript. D.J.B., K.N., R.H. and A.J.Q. supervised the project. All authors contributed to review and editing of the text.

## Competing interests
The authors declare no competing interests.
