## [Peer Review file · Communications Biology]

Single-Cell Transcriptomics Reveals the Molecular Basis of Human iPSC Cell Differentiation into Ectodermal Ocular Lineages

Corresponding Author: Professor Ryuhei Hayashi

Version 0:

Reviewer comments:

Reviewer #1

(Remarks to the Author)

Howard et al have generated a transcriptomic profile of human induced pluripotent stem cells (hiPSC) derived self-formed, ectodermal, autonomous, multi-zone (SEAM's) using single-cell library generation and sequencing technologies. The authors have identified the cellular identity in the developing four SEAM zones over a timeframe from Week 0-12. Gene expression analysis using 'FindAllMarker' function was applied to characterize the identity for each cell cluster. They observe that SEAM spontaneously differentiated and progressively represented the development of human eye by giving rise to multiple non ocular and ocular tissues (both surface ectodermal and neuroectodermal lineages). Overall, this is a carefully done study and the manuscript is well written. Specific comments and suggestions for improvement are listed below:

1. Introduction

- a. In first line (#57) Laura et al, have abbreviated SEAM as Self-formed, ectodermal, multi-zone. The 'autonomous' word in this abbreviation is missing.
- b. Is SEAM a 2D or the 3D generated structure? This needs to be stated in the Introduction.

2. General/Minor findings

- a. Sentences in some places are choppy and requires room for improvement to improve the overall readability.
- b. Spell and grammar check throughout the manuscript is needed.
- c. The subpanel of many figures has not been explained in the result section.
- d. Figure subpanels citation as parenthesis in result section is missing at many places throughout the manuscripts.
- e. Were the authors able to see the expression of opsins in the PR clusters of Week 12 SEAM? This needs to be discussed (in results/discussion section) to advance the utility of the SEAM models for studying inherited retinal degenerations.
- f. Data deposition and release in GEO (<https://www.ncbi.nlm.nih.gov/geo/>) upon publication is required and needs to be mentioned.

3. Experimental Procedures

- a. The Cat # of all the reagents utilized is missing. Eg: LN511E8, KGF, Y-27623, B-27, DMEM/F12, StemFit medium, etc.
- b. The authors mention about using human iPSC, clone 201B7 for the current study. Have the authors utilized this line for any other previous study? If so, citation is needed.
- c. What TC dishes/plates were utilized for SEAM cell culture. Pls indicate.
- d. Methodology of SEAM culture differentiation is described vaguely. This needs to be clearly written in detail under 'SEAM cell culture' section.
- e. A detailed composition & % of the supplement/growth factors utilized to make DM, CDM & CEM is missing. This could be included as supplemental table.
- f. How often were the cells fed during the SEAM cell culture? Was it every day or at different intervals for each medium?
- g. In Line # 508, authors mention about harvesting cells. A detail protocol of how harvesting was done needs to be explained and written here for the readers.
- h. The authors mention that the cells were harvested at 0-12 weeks. Was the collection done at every week in this period? This needs to be indicated based on what was done in this study. A justification for selecting a few time-points also needs to

be included and explained.

i. Was the experimental collection done in triplicate and then pooled for single cell library generation? What is the n=?

j. The abbreviated words like PC, pN, pK, DE needs to be elaborated during their first usage under 'single-cell data processing and analysis' section.

4. Results & Figures & Figure Legends

a. Fig 1

i. Line # 91 in results says about growing hiPSC for 10 days whereas in Line # 929, shows 40 days of iPSC culture. This is confusing and needs clarity.

ii. Line # 99 needs to refer Fig 1e in parenthesis for referring pluripotent markers.

iii. Line # 110 needs to refer Fig 1f in parenthesis for referring eye-field transcription factors figure.

iv. In Line # 929, authors need to indicate the type of SEAM image captured for 1a. For example, this could be written as – A typical 'phase contrast' SEAM image....

v. What does 'EFTFs' in Fig 1d mean? This needs to be abbreviated in the figure legend.

vi. In Line # 118, what kind of cellular identity is referred? Is it neural ectoderm and surface ectoderm? If so that needs to be specified. Additionally, the number of clusters identified in NE & SE from Fig 1h needs to be explained in result section.

vii. An explanation of results for cluster #10 from Fig 1i is needed and lacking. This specific cluster expresses pluripotent marker POU5F1, SE but do not express PAX6, DLX5 and BMP4.

viii. In Line # 119, POU5F1 could be mentioned in parenthesis after pluripotent marker to reflect the result in Fig 1i.

ix. Fig 1j shows the expression of FOXC2 and NGFR, which is not explained in the results section on Line # 138.

x. Similarly, Fig 1k shows the UMAP plots of SOX2 and OTX2 expression in SE and NE clusters which not explained in the results section on Line # 142.

xi. The authors also show the UMAP plots of POU5F1 expression mapped from the PSC cluster (#10) in the Fig 1k. This is not explained in the result section on Line #143.

xii. What does CNN in Line #936 mean? This needs to be abbreviated.

xiii. The type of representative image of SEAM in Fig 1c and 1g should be specified as 'phase contrast images' in their respective figure legends on Line # 931 and Line # 934.

b. Fig 2

i. In Line # 145, subtitle needs to be modified as week 2 and week 4. Writing as weeks 2-4 is inappropriate as authors don't show the results of week 3 SEAM in the result section.

ii. Is CRYAB gene shown in Fig 2C corresponds to early lens marker? An explanation for this is missing in result section in Line # 150-152 and in Figure legend- Line # 947.

iii. In Line # 170, the authors state as – The remaining four clusters at WK2.... Pls mention the cluster # for easy readers easy navigation.

iv. In Line # 172, include the name of the keratin family members corresponding to Fig 2f. Similarly in Line # 177, include the name of the Claudin genes referred in Fig 2f.

v. The type of representative image of SEAM in Fig 2a and 2g should be specified as 'phase contrast images' in their respective figure legends on Line # 945 and Line # 951.

vi. In Line # 191-193, the authors indicate the expression of DCX, TUBB3 and NEFM by cluster 4, 5 and 11 and cites Fig 2k. However, Fig 2k shows the violin plots of DCX and NEFL which doesn't match to the sentence written in result section. Is it NEFM or NEFL or both?

vii. In Line # 952-953 of Figure legend 2i, an explanation for SIX6 and SFRP2 plots are missing.

c. Fig 3

i. For Line # 234-235, where the authors describe about the robust expression of Pax6 expressed by Cluster 2 and 13 should cite Fig 3e to substantiate their statement.

ii. Cluster 4 in 3b is not described in the results.

iii. In the Figure Legend – Line # 962-963 for 3d, details of VSX1 and ATOH is missing.

iv. The bottom right figure of Fig 3e overlaps with a word (Epi), which needs to be removed for clear view of the image.

v. In the Figure Legend – Line # 964-965 for 3f, details of KRT4, MUC16, S100A9, GJB2 and GJB6 is missing.

vi. The authors provide the marker expression for clusters mapped in the Zone 2 (as represented in Fig 3h). However, in Line # 257-258, the authors do not give sufficient details about the cluster 3 & 9. What markers were expressed by these clusters?

vii. The marker expression details of NR (cluster # 0, 1 in Fig 3h) and RPC (cluster # 5 & 6 in Fig 3h) in the Zone 1 are not explained in the results (In the Line # 244-258).

viii. For Fig 3h - What is the difference between NR (Cluster # 0, 1 in Zone 2) and Neuron clusters (Cluster # 3, 9 in Zone 1)? Explain this in results.

ix. Line # 269-274, explains the cluster 14 expression by corresponding to Fig 3k. Few violin plots like PAX6 and SPRR2F shown in Fig 3k are not explained and missing in the results.

d. Fig 4

i. In the figure legend for Fig 4e – the violin plots of few genes like GJB2, GJB6, CXCL17, MUC1, MUC16, PAX6 is missing on Line # 978-980. Similar concerns with 4j noted, where the violin plots of few genes are not explained in figure legend on Line # 982-984.

ii. In the figure legend for Fig 4, abbreviation for ACs and RC is missing.

iii. In Fig 4c, the cell identity name (Neuron) of Cluster 16 is missing. The same is applicable to Fig 4h for cluster 15.

iv. In Fig 4c and 4h, check the spelling of RCG. This needs to be fixed on Line # 308.

e. Fig 5

i. In Fig 5b legend, on Line # 990, check the spelling of Pheatmap.

f. Fig 1-4 (General)

i. Labelling all the phase contrast images of SEAM in all figures (1-4) with zone numbers is recommended to track the

formation and changes of the multi-zones within SEAM during development.

Reviewer #2

(Remarks to the Author)

In this manuscript, Howard et al describe the single cell transcriptomic results for early stages of in vitro human ocular differentiation using their previously established protocol. Their goal was to identify molecular signatures of early human ocular development as well as to identify what other non-ocular cell types are present at these earlier stages of differentiation. The authors collected cells over a period of 12 weeks and performed molecular analyses using single cell RNAseq at time points ranging from week 0 to week 12 of differentiation. Similar work has been done by other groups, however, the work presented here follows and compares RNA signatures during the first few weeks of early ocular development.

Comments:

1. The authors state that “We have identified and molecularly characterized cellular populations which form within each of the concentric zones of the growing SEAM”. These zones are also marked in the culture images (where applicable), and more specifically in the UMAP plot in Figure 3h. However, it is not clear if the cells from each zone were isolated separately and then subjected to scRNAseq, or if the whole region was dissociated at the same time and the cell types in different zones are being assigned based on the signatures observed with scRNAseq. Please clarify.
2. It is possible that this reviewer could not find it but it appears that some of the cell-specific markers were not observed. For example, BEST1 and RPE65 were not observed even in late time point clusters. However, in several other differentiation protocols, these markers are observed by the latest time point tested here. Similarly, different research groups may employ different approaches for ocular/retinal differentiation. It would be beneficial to include some discussion on how these signatures may (or may not) vary depending on the approaches (and growth cocktails) used to generate ocular cells.
3. In figure 5, the authors show the pseudotime analysis and how the cells differentiate over time. However, it is not much useful as it shows a simple movement from PSCs to surface or neuroectoderm (which was concluded from data shown in previous figures). If possible, the following updates would help:
 - a. Figure 5d, please include the cell type instead of the cluster number in the right panel
 - b. This is not necessary, but will be helpful: Will it be possible to include another UMAP plot with clusters color-coded based on the time point (and not cell types)? So, essentially all cells from Wk 0 are colored blue, Week 1 are colored green etc. When such plot will be compared to Figure 5c, it could show how the composition of different cell types/lineages changes over time.
4. In some other differentiation approaches, forebrain development is also observed through anterior neuroectoderm. It is possible that none of such markers were observed in the scRNAseq analyses by the authors. Could the authors add some information on whether they observe any forebrain relevant cell types or not?
5. Methods sections needs more information. Since the whole manuscript is based on isolating cells at different time points during in vitro differentiation, it is important to, at least briefly, describe how the cells were isolated. Please also include the composition of the three different media used during SEAM differentiation (see comments 1 and 2).
6. Please check (and correct where applicable) the gene names in UMAP plots and feature plots (for example, RGCs are misspelled in feature plots).
7. Please include scale bars for inset images as well.

Reviewer #3

(Remarks to the Author)

Howard et al have generated a transcriptomic landscape of human induced pluripotent stem cells (hiPSC) derived self-formed, ectodermal, autonomous, multi-zone (SEAM's) using single-cell library generation and sequencing technologies. The authors have identified the cellular identity in the developing four SEAM zones over a timeframe from Week 0-12. Gene expression analysis using 'FindAllMarker' function was applied to recognize the cell identity for each cluster. Interestingly, SEAM spontaneously differentiated and progressively represented the development of human eye by giving rise to multiple non ocular and ocular tissues (both surface ectodermal and neuroectodermal lineages).

1. Introduction

- a. In first line (#57) Laura et al, have abbreviated SEAM as Self-formed, ectodermal, multi-zone. The 'autonomous' word in this abbreviation is missing.
- b. Is SEAM a 2D or the 3D generated structure? This needs to be stated in the Introduction.

2. General/Minor findings

- a. Sentences in some places are choppy and requires room for improvement to improve the overall readability.
- b. Spell and grammar check throughout the manuscript is needed.
- c. The subpanel of many figures has not been explained in the result section.
- d. Figure subpanels citation as parenthesis in result section is missing at many places throughout the manuscripts.
- e. Were the authors able to see the expression of opsins in the PR clusters of Week 12 SEAM? This needs to be discussed (in results/discussion section) to advance the utility of the SEAM models for studying inherited retinal degenerations.
- f. Data deposition and release in GEO (<https://www.ncbi.nlm.nih.gov/geo/>) upon publication is required and needs to be mentioned.

3. Experimental Procedures

- a. The Cat # of all the reagents utilized is missing. Eg: LN511E8, KGF, Y-27623, B-27, DMEM/F12, StemFit medium, etc.
- b. The authors mention about using human iPSC, clone 201B7 for the current study. Have the authors utilized this line for any other previous study? If so, citation is needed.
- c. What TC dishes/plates were utilized for SEAM cell culture. Pls indicate.
- d. Methodology of SEAM culture differentiation is described vaguely. This needs to be clearly written in detail under 'SEAM cell culture' section.
- e. A detailed composition & % of the supplement/growth factors utilized to make DM, CDM & CEM is missing. This could be included as supplemental table.
- f. How often were the cells fed during the SEAM cell culture? Was it every day or at different intervals for each medium?
- g. In Line # 508, authors mention about harvesting cells. A detail protocol of how harvesting was done needs to be explained and written here for the readers.
- h. The authors mention that the cells were harvested at 0-12 weeks. Was the collection done at every week in this period? This needs to be indicated based on what was done in this study. A justification for selecting a few time-points also needs to be included and explained.
- i. Was the experimental collection done in triplicate and then pooled for single cell library generation? What is the n=?
- j. The abbreviated words like PC, pN, pK, DE needs to be elaborated during their first usage under 'single-cell data processing and analysis' section.

4. Results & Figures & Figure Legends

a. Fig 1

- i. Line # 91 in results says about growing hiPSC for 10 days whereas in Line # 929, shows 40 days of iPSC culture. This is confusing and needs clarity.
- ii. Line # 99 needs to refer Fig 1e in parenthesis for referring pluripotent markers.
- iii. Line # 110 needs to refer Fig 1f in parenthesis for referring eye-field transcription factors figure.
- iv. In Line # 929, authors need to indicate the type of SEAM image captured for 1a. For example, this could be written as – A typical 'phase contrast' SEAM image....
- v. What does 'EFTFs' in Fig 1d mean? This needs to be abbreviated in the figure legend.
- vi. In Line # 118, what kind of cellular identity is referred? Is it neural ectoderm and surface ectoderm? If so that needs to be specified. Additionally, the number of clusters identified in NE & SE from Fig 1h needs to be explained in result section.
- vii. An explanation of results for cluster #10 from Fig 1i is needed and lacking. This specific cluster expresses pluripotent marker POU5F1, SE but do not express PAX6, DLX5 and BMP4.
- viii. In Line # 119, POU5F1 could be mentioned in parenthesis after pluripotent marker to reflect the result in Fig 1i.
- ix. Fig 1j shows the expression of FOXC2 and NGFR, which is not explained in the results section on Line # 138.
- x. Similarly, Fig 1k shows the UMAP plots of SOX2 and OTX2 expression in SE and NE clusters which not explained in the results section on Line # 142.
- xi. The authors also show the UMAP plots of POU5F1 expression mapped from the PSC cluster (#10) in the Fig 1k. This is not explained in the result section on Line #143.
- xii. What does CNN in Line #936 mean? This needs to be abbreviated.
- xiii. The type of representative image of SEAM in Fig 1c and 1g should be specified as 'phase contrast images' in their respective figure legends on Line # 931 and Line # 934.

b. Fig 2

- i. In Line # 145, subtitle needs to be modified as week 2 and week 4. Writing as weeks 2-4 is inappropriate as authors don't show the results of week 3 SEAM in the result section.
- ii. Is CRYAB gene shown in Fig 2C corresponds to early lens marker? An explanation for this is missing in result section in Line # 150-152 and in Figure legend- Line # 947.
- iii. In Line # 170, the authors state as – The remaining four clusters at WK2.... Pls mention the cluster # for easy readers easy navigation.
- iv. In Line # 172, include the name of the keratin family members corresponding to Fig 2f. Similarly in Line # 177, include the name of the Claudin genes referred in Fig 2f.
- v. The type of representative image of SEAM in Fig 2a and 2g should be specified as 'phase contrast images' in their respective figure legends on Line # 945 and Line # 951.
- vi. In Line # 191-193, the authors indicate the expression of DCX, TUBB3 and NEFM by cluster 4, 5 and 11 and cites Fig 2k. However, Fig 2k shows the violin plots of DCX and NEFL which doesn't match to the sentence written in result section. Is it NEFM or NEFL or both?
- vii. In Line # 952-953 of Figure legend 2i, an explanation for SIX6 and SFRP2 plots are missing.

c. Fig 3

- i. For Line # 234-235, where the authors describe about the robust expression of Pax6 expressed by Cluster 2 and 13 should cite Fig 3e to substantiate their statement.
- ii. Cluster 4 in 3b is not explained in the results.

- iii. In the Figure Legend – Line # 962-963 for 3d, details of VSX1 and ATOH is missing.
- iv. The bottom right figure of Fig 3e overlaps with a word (Epi), which needs to be removed for clear view of the image.
- v. In the Figure Legend – Line # 964-965 for 3f, details of KRT4, MUC16, S100A9, GJB2 and GJB6 is missing.
- vi. The authors provide the marker expression for clusters mapped in the Zone 2 (as represented in Fig 3h). However, in Line # 257-258, the authors do not give sufficient details about the cluster 3 & 9. What markers were expressed by these clusters?
- vii. The marker expression details of NR (cluster # 0, 1 in Fig 3h) and RPC (cluster # 5 & 6 in Fig 3h) in the Zone 1 are not explained in the results (In the Line # 244-258).
- viii. For Fig 3h - What is the difference between NR (Cluster # 0, 1 in Zone 2) and Neuron clusters (Cluster # 3, 9 in Zone 1)? Explain this in results.
- ix. Line # 269-274, explains the cluster 14 expression by corresponding to Fig 3k. Few violin plots like PAX6 and SPRR2F shown in Fig 3k are not explained and missing in the results.

d. Fig 4

- i. In the figure legend for Fig 4e – the violin plots of few genes like GJB2, GJB6, CXCL17, MUC1, MUC16, PAX6 is missing on Line # 978-980. Similar concerns with 4j where the violin plots of few genes are not explained in figure legend on Line # 982-984.
- ii. In the figure legend for Fig 4, abbreviation for ACs and RC is missing.
- iii. In Fig 4c, the cell identity name (Neuron) of Cluster 16 is missing. The same is applicable to Fig 4h for cluster 15.
- iv. In Fig 4c and 4h, check the spelling of RCG. This needs to be fixed on Line # 308.

e. Fig 5

- i. In Fig 5b legend, on Line # 990, check the spelling of Pheatmap.

f. Fig 1-4 (General)

- i. Labelling all the phase contrast images of SEAM in all figures (1-4) with zone #s is recommended to track the formation and changes of the multi-zones within SEAM during development.

Version 1:

Reviewer comments:

Reviewer #1

(Remarks to the Author)

The authors have addressed all my outstanding concerns. The manuscript is now suitable for publication.

Reviewer #2

(Remarks to the Author)

The authors have addressed most of the comments through their rebuttal and the revised manuscript. Comments on the revised manuscript are listed below:

1. One major question remains regarding the correlation of scRNAseq results with the zones observed by the authors. The authors in their rebuttal stated that "The scRNA-seq was performed on whole SEAMs". How did the authors then correlate the cell clusters with individual zones marked in the SEAM formation (for example, see lines 271-275)? Since the whole work is based on this approach, it is essential to detail and further explain this method. The current update in Methods section does not clearly explain the correlation.

2. The authors present data up to week 12, and do not observe mature photoreceptor marker expression. It could be because week 12 (D84) of differentiation is not too far along the developmental timeline. Around this time the early rod photoreceptor markers only start to show up in certain differentiation protocols. This point has been briefly discussed, however, adding more references would help. Additionally, it is also possible that the nature of differentiation (adherent culture) may be resulting in developmental limitations and thus lack of mature photoreceptor markers. Please include this possibility in discussion too.

Comment	Response	Revised Line #
Reviewer #1		
1. Introduction		
a. In first line (#57) Laura et al, have abbreviated SEAM as Self-formed, ectodermal, multi-zone. The 'autonomous' word in this abbreviation is missing.	This has been corrected in the revised text.	57
b. Is SEAM a 2D or 3D generated structure? This needs to be stated in the Introduction.	The SEAM is a 2D generated structure. The text has been amended in the revised manuscript to highlight this.	59
2. General/Minor findings		
a. Sentences in some places are choppy and require improvement to enhance overall readability.	We have revised sentences to improve readability where required.	
b. Spell and grammar check throughout the manuscript is needed.	Spell and grammar check completed throughout.	
c. The subpanel of many figures has not been explained in the results section.	We have edited the text throughout to include a description of all markers shown in figure subpanels.	
d. Figure subpanels citation as parenthesis in results section is missing in many places throughout the manuscript.	We have added subpanel citations in the results section where necessary.	
e. Were the authors able to see the expression of opsins in the PR clusters of Week 12 SEAM? This needs to be discussed (in results/discussion section) to advance the utility of the SEAM models for studying inherited retinal degenerations.	We did not detect expression of opsins in our data and discuss this in lines 489-496.	489-496
f. Data deposition and release in GEO (https://www.ncbi.nlm.nih.gov/geo/) upon publication is required.	Datasets have been deposited in NCBI GEO under accession number GSE263987.	630
3. Experimental Procedures		
a. The Cat # of all the reagents utilized is missing. Eg: LNSLIES, KGF, -27623, B-27, DMEM/F12, Stemfit medium, etc.	Catalogue numbers have been included in the text throughout the methods section.	565-599

b. The authors mention using human iPSC, clone 20187 for the current study. Have the authors utilized this line for any other previous study? If so, citation is needed.	This hiPSC line has been used in a number of published studies and is also described in 'Takahashi K, et al. Induction of pluripotent stem cells from adult human fibroblasts by defined factors. Cell 131, 861-872 (2007)'. Clone 201B7 is one of the first reported clones from Shinya Yamanaka's lab and is globally distributed and used in many labs. We have updated the in-text citations to reflect this.	566
c. What TC dishes/plates were utilized for SEAM cell culture? Please indicate.	We use 6 well culture plates (353046, Corning, NY, USA) coated with 0.5 µg/cm² LN511E8. The methods section has been updated to include this information.	571-572
d. Methodology of SEAM culture differentiation is described vaguely. This needs to be clearly written in detail under 'SEAM cell culture' section.	This section has now been significantly expanded to include these experimental details.	565-582
e. A detailed composition & percentage of the supplement/growth factors utilized to make OM, CDM & CEM is missing. This could be included as a supplemental table.	This information is now provided in a new Supplementary Table 1.	581
f. How often were the cells fed during the SEAM cell culture? Was it every day or at different intervals for each medium?	Medium changes were performed once every 2-3 days (three times per week). This information has been included in the revised methods section.	580-581
g. In Line #508, authors mention about harvesting cells. A detailed protocol of how harvesting was done needs to be explained and written here for the readers.	This section has now been significantly expanded to include these experimental details.	584-593
h. The authors mention that the cells were harvested at 0-12 weeks. Was the collection done at every week in this period? This needs to be indicated based on what was done in this study. A justification for selecting a few time-points also needs to be included and explained.	Collection was performed at the 8 timepoints indicated in the manuscript (clarified on lines 584-586). We have previously ascertained that these intervals provide a sufficiently accurate depiction of the different stages of SEAM formation and maturation. We essentially analyse every two weeks during the culture period (which at 0, 4 and 8 weeks coincides with the point of media composition change) and also include the WK1 sample in order to capture dramatic changes which may occur at the very early stages of differentiation.	584-587

i. Was the experimental collection done in triplicate and then pooled for single cell library generation? What is the n=?	Libraries were prepared from cells from individual wells for each of the eight timepoints in the same single series experiment as n=1. This is now clarified on line 587. Though we recognise that SEAM culture is a robust system, success of SEAM culture, reproducibility, and reliability of results was assessed by i) the formation of characteristic zones, ii) the timings of appearance of the main ocular cell types being consistent with those reported previously (Hayashi et al., 2016; 2017) and iii) the consistency of gene expression patterns across 8 separate developmental stages and across many thousands of individual cells.	587-588
j. The abbreviated words like PC, pN, pk, DE need to be elaborated during their first usage under 'single-cell data processing and analysis' section.	These have been elaborated in the 'single-cell data processing and analysis' section.	608-610
		
Results & Figures & Figure legends		
a. Fig 1		
i. Line #91 in results says about growing hiPSC for 10 days whereas in Line #929, it shows 40 days of IPSC culture. This is confusing and needs clarity.	SEAMs are grown in Stemfit medium to preserve pluripotency before the medium is changed to differentiation medium which promotes SEAM formation. The figure legend has been changed to clarify that the image shows a differentiating SEAM.	1098-1099
ii. Line #99 needs to refer to Fig 1e in parenthesis for referring pluripotent markers.	'Fig. 1e' has been added to the revised text.	100
iii. Line #110 needs to refer to Fig 1f in parenthesis for referring eye-field transcription factors figure.	The EFTFs referred to here are from the literature. However, 'Fig. 1d' has been added to refer the reader to the numerical clusters.	112
iv. In Line #929, authors need to indicate the type of SEAM image captured for 1a. For example, this could be written as – 'A typical phase-contrast SEAM image'.	The text has been updated to indicate that a phase-contrast image is shown.	1098
v. What does 'EFTFs' in Fig. 1d mean? This need to be abbreviated in the figure legend	'EFTFs – eye-field transcription factors' – has been included in the figure legend.	1111
vi. In Line # 118, what kind of cellular identity is referred? Is it neural ectoderm and surface ectoderm? If	The divergence of cellular identity towards either surface ectodermal or neuroectodermal fates is now indicated	119-120

so that needs to be specified. Additionally, the number of clusters identified in NE & SE from Fig 1h needs to be explained in the result section	in the text and cluster numbers are provided. We have updated the figure labelling to highlight this and to allow direct comparison with Seurat cluster numbering.	
vii. An explanation of results for cluster #10 from Fig 1i is needed and lacking. This specific cluster expresses pluripotent marker POU5F1, SE but does not express PAX6, DLX5 and BMP4.	Cells in cluster 10 are PAX6 -negative and are likely pluripotent cells which are newly differentiating towards surface ectodermal fates, but before the establishment of ocular vs. non-ocular commitment. This is now discussed in the revised text.	130-132
viii. In Line # 119, POU5F1 could be mentioned in parenthesis after pluripotent marker to reflect the result in Fig 1i.	This has been included in the revised text.	121
ix. Fig 1j shows the expression of FOXC2 and NGFR, which is not explained in the results section on Line # 138.	We have now referred to these markers in the text of the results section.	143
x. Similarly, Fig 1k shows the UMAP plots of SOX2 and OTX2 expression in SE and NE clusters which are not explained in the results section on Line # 142.	We have now referred to these markers in the text of the results section.	146-149
xi. The authors also show the UMAP plots of POU5F1 expression mapped from the PSC cluster (#10) in Fig 1k. This is not explained in the results section on Line # 143.	Cells in cluster 10 are POU5F1 -positive and PAX6 -negative and are likely pluripotent cells which are newly differentiating towards surface ectodermal fates. This is now discussed in the revised text	130-132
xii. What does CNN in Line #936 mean? This needs to be abbreviated.	'CNN – cranial neural crest' has been added to the revised text.	1114
xiii. The type of representative image of SEAM in Fig 1c and 1g should be specified as 'phase contrast images' in their respective figure legends on Line # 931 and Line # 934.	The text has been updated to indicate that phase-contrast images are shown.	1101, 1104
b. Fig 2		
i. In Line # 145, subtitle needs to be modified as week 2 and week 4. Writing as weeks 2-4 is inappropriate as authors don't show the results of week 3 SEAM in the result section.	This has been updated to indicate that weeks 2 AND 4 are shown.	153
ii. Is CRYAB gene shown in Fig 2C corresponds to early lens marker? An explanation for this is missing in result section in Line # 150-152 and in Figure legend - Line # 947.	We have referred to and discussed the expression of CRYAB in the revised text.	160, 1119

iii. In Line # 170, the authors state as – The remaining four clusters at WK2.... Pls mention the cluster # for easy readers navigation.	Cluster numbers have been included in the revised text.	189
iv. In Line # 172, include the name of the keratin family members corresponding to Fig 2f. Similarly in Line # 177, include the name of the Claudin genes referred in Fig 2f.	We have updated the text to include specific marker names.	191, 202
v. The type of representative image of SEAM in Fig 2a and 2g should be specified as ‘phase contrast images’ in their respective figure legends on Line # 945 and Line # 951.	The text has been updated to indicate that phase-contrast images are shown.	1117, 1122
vi. In Line # 191-193, the authors indicate the expression of DCX, TUBB3 and NEFM by cluster 4, 5 and 11 and cites Fig 2k. However, Fig 2k shows the violin plots of DCX and NEFL which doesn’t match to the sentence written in result section. Is it NEFM or NEFL or both?	Figure 2k has been updated to include TUBB3, DCX, NEFM and NEFL. We also refer to this in the relevant results sections of the text.	216
vii. In Line # 952-953 of Figure legend 2i, an explanation for SIX6 and SFRP2 plots are missing.	This information has been added to the revised text.	1129
c. Fig 3		
i. For Line # 234-235, where the authors describe about the robust expression of Pax6 expressed by Cluster 2 and 13 should cite Fig 3e to substantiate their statement.	Fig. 3e is cited in the revised text.	264
ii. Cluster 4 in 3b is not described in the results.	Cluster 4 is described in the revised text.	234-235
iii. In the Figure Legend – Line # 962-963 for 3d, details of VSX1 and ATOH is missing.	This information has been added to the revised text.	1141
iv. The bottom right figure of Fig 3e overlaps with a word (Epi), which needs to be removed for clear view of the image.	This label has been removed from Fig. 3e	
v. In the Figure Legend – Line # 964-965 for 3f, details of KRT4, MUC16, S100A9, GJB2 and GJB6 is missing.	This information has been added to the revised text.	1143-1144
vi. The authors provide the marker expression for clusters mapped in the Zone 2 (as represented in Fig 3h). However, in Line # 257-258, the authors do not give sufficient details about the cluster 3 & 9. What	We have described clusters 3 and 9 and their relevance to the zoned SEAM model in the revised text.	289-293

markers were expressed by these clusters?		
vii. The marker expression details of NR (cluster # 0, 1 in Fig 3h) and RPC (cluster # 5 & 6 in Fig 3h) in the Zone 1 are not explained in the results (In the Line # 244-258).	This has been clarified in the text.	276-278
viii. For Fig 3h - What is the difference between NR (Cluster # 0, 1 in Zone 2) and Neuron clusters (Cluster # 3, 9 in Zone 1)? Explain this in results.	We have discussed the neuron clusters & their relationship to the NR in the revised text.	289-293
ix. Line # 269-274, explains the cluster 14 expression by corresponding to Fig 3k. Few violin plots like PAX6 and SPRR2F shown in Fig 3k are not explained and missing in the results.	This information has been added to the revised text.	305
d. Fig 4		
i. In the figure legend for Fig 4e – the violin plots of few genes like GJB2, GJB6, CXCL17, MUC1, MUC16, PAX6 is missing on Line # 978-980. Similar concerns with 4j noted, where the violin plots of few genes are not explained in figure legend on Line # 982-984.	This information has been added to the revised text.	1162-1163 & 1167-1168
ii. In the figure legend for Fig 4, abbreviation for ACs and RC is missing.	This has been added to the Fig. 4 legend.	1172
iii. In Fig 4c, the cell identity name (Neuron) of Cluster 16 is missing. The same is applicable to Fig 4h for cluster 15.	Corrected in both figures.	
iv. In Fig 4c and 4h, check the spelling of RCG. This needs to be fixed on Line # 308.	Corrected for both.	320, 355
e. Fig 5		
i. In Fig 5b legend, on Line # 990, check the spelling of Pheatmap.	This is correct, but text has been changed to indicate that 'pheatmap' is a type of heatmap produced in R.	1178
f. Fig 1-4 (General)		

i. Labelling all the phase contrast images of SEAM in all figures (1-4) with zone numbers is recommended to track the formation and changes of the multi-zones within SEAM during development.	This is difficult to accurately define in the earlier SEAMs, however, we have added labelling where possible.	
Reviewer #2		
1. The authors state that “We have identified and molecularly characterized cellular populations which form within each of the concentric zones of the growing SEAM”. These zones are also marked in the culture images (where applicable), and more specifically in the UMAP plot in Figure 3h. However, it is not clear if the cells from each zone were isolated separately and then subjected to scRNAseq, or if the whole region was dissociated at the same time and the cell types in different zones are being assigned based on the signatures observed with scRNAseq. Please clarify.	scRNA-seq was performed on whole SEAMs. This has been clarified in the text.	584-593
2. It is possible that this reviewer could not find it but it appears that some of the cell-specific markers were not observed. For example, BEST1 and RPE65 were not observed even in late time point clusters. However, in several other differentiation protocols, these markers are observed by the latest time point tested here. Similarly, different research groups may employ different approaches for ocular/retinal differentiation. It would be beneficial to include some discussion on how these signatures may (or may not) vary depending on the approaches (and growth cocktails) used to generate ocular cells.	We have now included results which show the expression of BEST1 and RPE65 (together with other RPE markers) in WK12 SEAMs in a new Supplementary Fig. 7, together with reasoning for lack of expression of some mature markers in the RPE and photoreceptor clusters. We have also acknowledged how differences in culture conditions might account for this and cite other relevant studies.	351-354, Figure S7, 483-489

3. In figure 5, the authors show the pseudotime analysis and how the cells differentiate over time. However, it is not much useful as it shows a simple movement from PSCs to surface or neuroectoderm (which was concluded from data shown in previous figures). If possible, the following updates would help: a. Figure 5d, please include the cell type instead of the cluster number in the right panel b. This is not necessary, but will be helpful: Will it be possible to include another UMAP plot with clusters color-coded based on the time point (and not cell types)? So, essentially all cells from Wk 0 are colored blue, Week 1 are colored green etc. When such plot will be compared to Figure 5c, it could show how the composition of different cell types/lineages changes over time.	a) Cell type annotations have now been included in Fig. 5d b) UMAP plots with clusters color-coded based on time points are included in Supplementary Figure 9a, together with a direct comparison between time point and Seurat cluster. This is referred to in the revised text.	384
4. In some other differentiation approaches, forebrain development is also observed through anterior neuroectoderm. It is possible that none of such markers were observed in the scRNAseq analyses by the authors. Could the authors add some information on whether they observe any forebrain relevant cell types or not?	A number of shared markers link anterior neuroectoderm with forebrain and early eye development. However, we have included new expression data for FOXP1, which is crucial for forebrain development, in Fig. 2e and Supplementary Fig. 6b, and discuss this in the revised manuscript.	180-181, Fig. 2e, Supplementary Fig. 6b, 291-293
5. Methods sections needs more information. Since the whole manuscript is based on isolating cells at different time points during in vitro differentiation, it is important to, at least briefly, describe how the cells were isolated. Please also include the composition of the three different media used during SEAM differentiation (see comments 1 and 2).	This section has now been significantly expanded to include these experimental details. We have also included media composition details in a new Supplementary Table 1.	564-599
6. Please check (and correct where applicable) the gene names in UMAP plots and feature plots (for example, RGCs are misspelled in feature plots).	This has been corrected where necessary.	

7. Please include scale bars for inset images as well.	Scale bars have been added to inset images and are referred to in the relevant legends.	
--	---	--

OUTSTANDING REVIEWERS' COMMENTS:

Reviewer #2 (Remarks to the Author):

The authors have addressed most of the comments through their rebuttal and the revised manuscript. Comments on the revised manuscript are listed below:

1. One major question remains regarding the correlation of scRNAseq results with the zones observed by the authors. The authors in their rebuttal stated that "The scRNA-seq was performed on whole SEAMs". How did the authors then correlate the cell clusters with individual zones marked in the SEAM formation...?

Annotated clusters were correlated with respective SEAM zones by comparing single-cell expression data with established SEAM zone markers and cell phenotypes. This has been clarified in the new revised text (with mark-up) in lines 574-577.

2. The authors present data up to week 12, and do not observe mature photoreceptor marker expression. It could be because week 12 (D84) of differentiation is not too far along the developmental timeline. Around this time the early rod photoreceptor markers only start to show up in certain differentiation protocols. This point has been briefly discussed, however, adding more references would help. Additionally, it is also possible that the nature of differentiation (adherent culture) may be resulting in developmental limitations and thus lack of mature photoreceptor markers. Please include this possibility in discussion too.

It is highly likely that the absence of mature photoreceptor markers in our data is a result of developmental timing, and that this expression is simply not detectable during the period covered by our study. We have cited additional studies which support this. We have also considered that our adherent culture may influence differentiation, and discuss this in lines 451-463 of the newly revised manuscript.